ecology, health and disease and epidemiology

COVID-19, coronavirus, SARS-CoV-2, non-pharmaceutical interventions

**Author for correspondence:**
John M. Drake
e-mail: jdrake@uga.edu

# Five approaches to the suppression of SARS-CoV-2 without intensive social distancing

John M. Drake[1,2], Kyle Dahlin[1,2], Pejman Rohani[1,2] and Andreas Handel[2,3]

[1]Odum School of Ecology, [2]Center for the Ecology of Infectious Diseases, and [3]College of Public Health, Epidemiology and Biostatistics, University of Georgia, Athens, GA 30602, USA

JMD, 0000-0003-4646-1235; KD, 0000-0002-1116-8074; PR, 0000-0002-7221-3801; AH, 0000-0002-4622-1146

Initial efforts to mitigate transmission of SARS-CoV-2 relied on intensive social distancing measures such as school and workplace closures, shelter-in-place orders and prohibitions on the gathering of people. Other non-pharmaceutical interventions for suppressing transmission include active case finding, contact tracing, quarantine, immunity or health certification, and a wide range of personal protective measures. Here we investigate the potential effectiveness of these alternative approaches to suppression. We introduce a conceptual framework represented by two mathematical models that differ in strategy. We find both strategies may be effective, although both require extensive testing and work within a relatively narrow range of conditions. Generalized protective measures such as wearing face masks, improved hygiene and local reductions in density are found to significantly increase the effectiveness of targeted interventions.

## 1. Introduction

Efforts to control the COVID-19 pandemic have resulted in unprecedented economic impacts. The path to 'reopening' the economy will require strategies for suppressing transmission of SARS-CoV-2 that do not depend exclusively on stringent interventions and such intensive social distancing policies as school and workplace closure and mandatory shelter-in-place (i.e. 'lockdowns'). Several different approaches to suppressing transmission have been suggested [1–4], but there has been little systematic comparison of the effectiveness, cost or robustness of these strategies [5]. We developed models for five approaches to suppressing transmission without the need for completely eliminating personal and business activities. These models illustrate the similarities and differences among these approaches and help to identify their distinctive strengths and weaknesses.

Our conceptual framework distinguishes between *targeted* and *generalized* non-pharmaceutical interventions (NPIs). Targeted interventions are interventions that are applied to specifically identified individuals in a population, typically based on infection or exposure status. The effect of various targeted interventions has been considered in several previous works: isolation and contact tracing [2], quarantine and symptom monitoring [6], travel restrictions [7,8], contact tracing and household quarantine [9], targeted protections for vulnerable populations or 'cocooning' [10]. Generalized interventions are behavioural or environmental interventions that are adopted broadly within a population. Past modelling studies have also considered the effects of various generalized interventions including: physical distancing [11], school closures, physical distancing, shielding of elderly, self-isolation and lockdown [12], and physical distancing and mask use [13]. Further studies have considered combinations of targeted and generalized interventions: physical distancing with

contact tracing [14], isolation, testing, contact tracing and physical distancing [5], individual quarantine and active contact monitoring [15].

We consider four targeted interventions that belong to two distinct strategies. The strategies are structurally different in the sense that their flow diagram representations are incommensurate. To our knowledge, no other studies have considered and compared the effectiveness, cost or robustness of such strategies.

## (a) Strategy 1: targeting infected persons

The first strategy targets infected people to limit transmission risk. Each approach in this strategy represents an escalation of intervention.

1. *Active case finding.* Active case finding refers to all efforts that actively seek to identify cases, for instance, by testing of healthcare workers and others who may have high occupational exposures, testing contacts of cases and adopting minimally exclusive testing criteria. It is assumed that identified cases are isolated and that onward transmission is eliminated or greatly reduced upon isolation. Basically, we are equating active case finding to widespread testing. Active case finding contrasts with passive case finding, which we define as the detection of cases among symptomatic patients who present to medical services for diagnosis of symptoms and receive a test only after meeting some criteria.

2. *Contact tracing.* Contact tracing is the identification, communication with and monitoring of possible exposures of known cases. Contact tracing increases awareness among the subset of the population most likely to develop symptoms, decreases transmission from traced contacts who are encouraged to isolate, and increases the rate of case finding in the population. Contact tracing may be performed by interviewing cases or family members of cases or with technological aids like mobile phone apps [3]. Prior to the COVID-19 pandemic, contact tracing had never been attempted at the scale that would be required to be effective in suppressing SARS-CoV-2 and several studies have considered how such scale-up might be accomplished [2–4].

3. *Quarantine.* Quarantine represents an escalation of intervention severity that amplifies the impact of contact tracing. This approach involves isolating traced contacts to the same degree that known cases are isolated. The major effect of this approach is that it reduces the dependence on finding secondary cases (because secondary cases are already identified as contacts) and reduces or eliminates onward transmission from these cases (because the case is already in isolation when symptoms begin). Another effect is that it reduces the average contact rate within the population. Effectively, the portion of the population that is in quarantine is engaged in intensive social distancing, which can be thought of as a 'partial lockdown' that is tunable based on the intensity of contact tracing.

## (b) Strategy 2: targeting uninfected persons

The second strategy comprises one approach targeting healthy people to limit exposure.

4. *Certification.* Certification is an approach that relaxes social distancing in stages. Under this approach, individuals are certified to be infection free before returning to daily routines such as school, work and shopping. Certification can be *durable* (valid for an extended period of time, for instance based on an antibody test) or *temporary* (valid for a short period of time, for instance because one has recently tested negative by RNA test). Durable certification, in our model, does not lead to a reduction in transmission, but may be essential for the provision of essential goods and services during periods of high transmission. This is similar to the 'shield immunity' concept of Weitz *et al.* [16]. To our knowledge, the effectiveness of a policy of temporary certification has not been evaluated in other modelling studies.

We note that these strategies have different political, philosophical, ethical and behavioural implications. For instance, Strategy 1 may disincentivize care-seeking because receiving a positive test could preclude one from working, whereas Strategy 2 may incentivize care-seeking because a negative diagnostic test or positive antibody test is required to work. Similarly, Strategy 1 prioritizes a right to work whereas Strategy 2 prioritizes a duty to protect. In addition, Strategy 1 and Strategy 2 approaches could be combined. However, because they are structurally different, we do not consider such combinations here.

## (c) Generalized interventions

In addition, these targeted interventions may be used in combination with generalized interventions. Generalized interventions act by reducing transmission or exposure broadly in a population and are not structurally different to the targeted strategies they are combined with in the sense that they may be added to either targeted strategy without modifying the topology of the flow diagram.

5. *Generalized interventions.* Generalized interventions are behavioural or environmental interventions that are adopted broadly within a population, including: wearing face masks [17]; improved hand hygiene [18,19]; improved cleaning and disinfection of surfaces [19,20]; greater provision of sick leave and increased enforcement of school and workplace guidelines for staying home when sick [21]; contactless transactions [22]; use of infection barriers in shops, restaurants, and waiting areas; distribution of hand sanitizer in public places; behavioural change (e.g. elbow/fist bump versus handshake [23]); use of personal rather than public transport; micro-social-distancing (e.g. limiting physical contact, queue spacing); and public policies that limit local aggregations of people such as limits on the number of people allowed in a shop and disallowing large events.

## (d) Overview

Below, we present conceptual models devised to be realistic for SARS-CoV-2, but they are not fit to data from any particular population. We studied the dynamics of active case finding, contact tracing, quarantine, and certification individually and in combination with generalized interventions after a 'first wave' that infects a small fraction of the population. For comparison, we also consider the two limiting cases of maintaining intensive social distancing and doing

nothing. The models are parametrized for a population of 10 million people, slightly larger than London (8.9 million) and New York City (8.3 million), and slightly smaller than the US state of Georgia (10.6 million), but they may be parametrized for a population of any size.

We use the models to answer a number of general strategic questions about these five approaches to suppressing transmission without social distancing.

1. How much might generalized interventions (without targeted interventions) reduce the total outbreak size compared with reference scenarios?
2. When are contact tracing and quarantine most beneficial?
3. What benefit does quarantine add to contact tracing?
4. When can certification be effective?
5. How does the extent of presymptomatic transmission affect the choice of intervention strategy?

## 2. Methods

Our approach uses models characterizing the two over-arching strategies of non-pharmaceutical interventions described above: active case finding and certification. From these we obtain two primary quantities of interest for each strategy: outbreak size over a 3-year horizon and the control reproduction number. Estimates of outbreak size are used to provide a quantity for direct comparison of the impact of strategies and the intensity at which they are applied. The control reproduction number, $R_c$, is a threshold quantity which informs how control parameters may reduce the potential for a substantial outbreak. Reproduction numbers generally represent the average number of new cases in a population induced by a single infectious individual. If a control scheme reduces $R_c$ below one, then the transmission of the pathogen will be substantially reduced.

### (a) Strategy 1: Active case finding, contact tracing and quarantine

The system of equations for Strategy 1 is

$$\dot{S}_u = \kappa S_t - \alpha S_u I_t - \beta(I_u + b_{I_t}I_t + b_{L_u}L_u + b_{L_t}L_t)S_u, \quad (1.1a)$$

$$\dot{L}_u = \beta(I_u + b_{I_t}I_t + b_{L_u}L_u + b_{L_t}L_t)S_u - \alpha L_u I_t - \sigma L_u, \quad (1.1b)$$

$$\dot{I}_u = (1-q)\sigma L_u - \gamma I_u, \quad (1.1c)$$

$$\dot{S}_t = \begin{cases} \alpha S_u I_t - \kappa S_t - \beta(0 I_u + b_{I_t}I_t + 0 L_u + b_{L_t}L_t)S_t, \\ \qquad \text{if in quarantine mode} \\ \alpha S_u I_t - \kappa S_t - \beta(I_u + b_{I_t}I_t + b_{L_u}L_u + b_{L_t}L_t)S_t, \quad \text{otherwise,} \end{cases} \quad (1.1d)$$

$$\dot{L}_t = \begin{cases} \beta(0 I_u + b_{I_t}I_t + 0 L_u + b_{L_t}L_t)S_t + \alpha L_u I_t - \sigma L_t, \\ \qquad \text{if in quarantine mode} \\ \beta(I_u + b_{I_t}I_t + b_{L_u}L_u + b_{L_t}L_t)S_t + \alpha L_u I_t - \sigma L_t, \quad \text{otherwise,} \end{cases} \quad (1.1e)$$

$$\dot{I}_t = \sigma(L_t + q L_u) - \gamma I_t \quad (1.1f)$$

and $\quad \dot{R} = \gamma(I_u + I_t). \quad (1.1g)$

This model (figure 1) supposes that there are both traced (labelled with the subscript $t$) and untraced persons (subscript $u$) who are susceptible, incubating and fully infectious as well as one pool of recovered and removed, designated $S$, $L$, $I$ and $R$, respectively. In contrast to the usual convention (where incubating cases are considered to be 'exposed', designated $E$), as presymptomatic transmission is well documented and

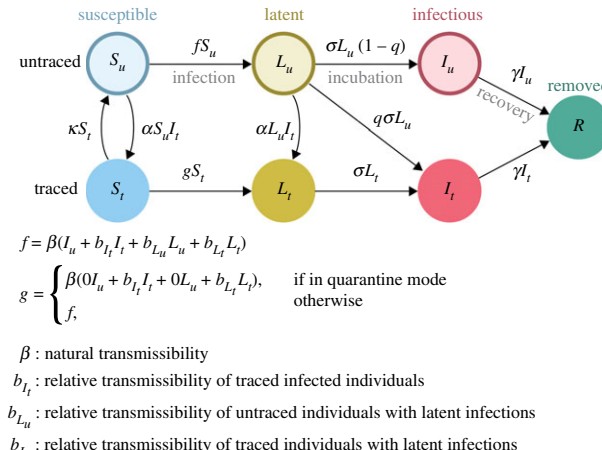

$f = \beta(I_u + b_{I_t}I_t + b_{L_u}L_u + b_{L_t}L_t)$

$g = \begin{cases} \beta(0 I_u + b_{I_t}I_t + 0 L_u + b_{L_t}L_t), & \text{if in quarantine mode} \\ f, & \text{otherwise} \end{cases}$

$\beta$ : natural transmissibility

$b_{I_t}$ : relative transmissibility of traced infected individuals

$b_{L_u}$ : relative transmissibility of untraced individuals with latent infections

$b_{L_t}$ : relative transmissibility of traced individuals with latent infections

$q$ : probability of case detection

$1/\sigma$ : average time in $L$ class $\qquad 1/\kappa$ : average time to receive RNA test

$1/\gamma$ : average time in $I$ class $\qquad \alpha$ : speed of contact identification

**Figure 1.** Compartmental model for Strategy 1 interventions. (Online version in colour.)

possibly quite important in the context of COVID-19 interventions [24–26], our model replaces $E$ with $L$ (for 'latent'). These latent or incubating infection may contribute to the force of infection, at an intensity reduced by a factor of $b_L$. The $I$ compartments comprise all fully infectious individuals, both symptomatic and asymptomatic. We assume that asymptomatic individuals are as capable of transmitting SARS-CoV-2 onwards, but that these cases are more difficult to detect (represented by the case detection probability, $q$). This difficulty may arise because asymptomatic individuals are less likely to seek testing or because they may have viraemia below the detection threshold of the test. The compartments represent the true epidemiological status of individuals while contact tracing and RNA testing are considered to be imperfect methods of determining the status of individuals. Thus, for example, susceptible and latent individuals obtain sero-negative results if they receive an RNA test. The factors $b_{I_t}$, $b_{L_u}$ and $b_{L_t}$ correspond to the differential rates of infection due to individuals in those respective compartments measured relative to the transmission rate of untraced infectious individuals. Transmission is assumed to occur through mass action.

Untraced susceptible individuals ($S_u$) may develop a latent infection due to contact with (traced or untraced) infected or latently infected individuals at the rate $\beta(I_u + b_{I_t}I_t + b_{L_u}L_u + b_{L_t}L_t)$ (1.1a). If associated with a traced infected individual, untraced susceptible individuals move to the traced susceptible compartment at the rate $\alpha I_t$. Traced susceptible individuals can receive an RNA test and move to the untraced susceptible compartment at the rate $\kappa$ in order to leave quarantine. Untraced latently infected individuals ($L_u$) may become traced due to association with a traced infected individual at rate $\alpha I_t$ or become symptomatic at rate $\sigma$ (1.1b). Upon displaying symptoms, untraced latent individuals are assumed to enter isolation with the empirically observed case ascertainment probability of $q \leq 1$. Case ascertainment reflects the combined effects of both passive case finding ($q_p$) and active case finding ($q_a$) that may include less than 100% of known cases so that $q = q_p + q_a$ (1.1c). Incomplete case finding could be either intentional or unintentional. All symptomatic individuals recover at the rate $\gamma$.

Proc. R. Soc. B 288: 20203074

The rate at which traced susceptible individuals ($S_t$) may become infected is dependent on whether or not quarantine is in effect. If quarantined, traced susceptible individuals may be infected at the reduced rate $\beta b_{I_t} I_t + \beta b_{L_t} L_t$ and if not quarantined, they are infected at the same rate as untraced susceptible individuals (1.1d). At rate $\kappa$, traced susceptible individuals ($S_t$) receive an RNA test and move to the untraced susceptible compartment. Traced latently infected individuals develop symptoms at rate $\sigma$ (1.1e). It is assumed that all traced latent individuals remain in the programme as new cases upon the development of symptoms. Traced symptomatic individuals ($I_t$) recover at rate $\gamma$. Finally, the recovered and removed compartment ($R$) is made up of all traced and untraced symptomatic individuals which have recovered (1.1g).

Using the next-generation matrix [27–29], we obtained the following expression for the basic reproduction number of this model, in the absence of active case finding ($q_a = 0$ and $q = q_p$):

$$R_0 = \frac{\tilde{\beta} b_{L_u}}{\sigma} + \frac{\tilde{\beta}(1 - q_p)}{\gamma} + \frac{\tilde{\beta} b_{I_t} q_p}{\gamma}, \tag{1.2}$$

where $\tilde{\beta} = \beta N$ with $N$ the initial total population size. With active case finding ($q_a > 0$), we obtain a control reproduction number:

$$R_c = \frac{\tilde{\beta} b_{L_u}}{\sigma} + \frac{\tilde{\beta}(1 - q)}{\gamma} + \frac{\tilde{\beta} b_{I_t} q}{\gamma} = R_0 - q_a \frac{\tilde{\beta}}{\gamma}(1 - b_{I_t}). \tag{1.3}$$

Note that this quantity will be less than $R_0$ since traced infected individuals are considered to be relatively less capable of transmission ($b_{I_t} < 1$). Thus, overall, $R_c$ is a weighted sum of transmission contributions from the incubating, untraced and traced infectious individuals, respectively. It may sometimes be useful to have the critical case finding value $q^*$ at which $R_c = 1$. We can determine this to be

$$q^* = \frac{\tilde{\beta} b_{L_u} \gamma + \tilde{\beta} \sigma - \sigma \gamma}{\tilde{\beta} \sigma - \tilde{\beta} b_{I_t} \sigma}. \tag{1.4}$$

Accordingly, for case ascertainment to be able to prevent an epidemic, we must have that $q^* < 1$. For a given set of parameters, this may not be possible. In general, if $R_c$ exceeds one when there is perfect case ascertainment ($q_a = 1$), then the critical case finding value is not defined. Specifically, if the contribution of the latent class to the reproduction number exceeds one ($\tilde{\beta} b_{L_u}/\sigma > 1$), then case ascertainment cannot prevent an outbreak.

Care should be taken in the interpretation of the force of infection functions $f$ and $g$. The force of infection is formulated such that a 'natural' transmissibility $\beta$, assumed to represent the baseline contagiousness of an untraced symptomatic case circulating in the population, is multiplied by a factor ($b_{I_t} < 1$, $b_{L_u} < 1$, or $b_{L_t} < 1$) to represent the contagiousness of latent infections and isolated cases. This allows that infection from traced and untraced individuals may occur at different rates and thus we think of the transmissibility 'attaching' to the class of the infected individual (traced or untraced). *Completely effective* isolation is represented by setting $b_{L_t} = 0$ and $b_{I_t} = 0$. *Active case finding* is represented by setting $\alpha = 0$ and $\kappa = 0$ and tuning $q$ to represent different levels of active case finding. *Quarantine* is represented by setting $g = \beta(0 + b_{I_t} I_t + 0 + b_{L_t} L_t)$ and setting $b_{I_t}$ and $b_{L_t}$ to values that reflect the amount of transmission that may happen within a household where a person is quarantined. Completely effective quarantine is represented by setting $g = 0$, $b_{L_t} = 0$ and $b_{I_t} = 0$. This model reduces to the

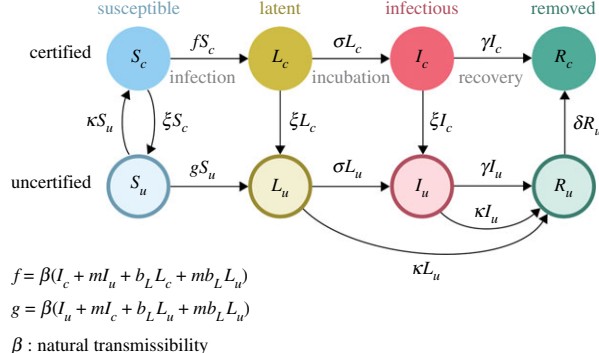

$f = \beta(I_c + mI_u + b_L L_c + mb_L L_u)$

$g = \beta(I_u + mI_c + b_L L_u + mb_L L_u)$

$\beta$ : natural transmissibility

$m$ : inter-class mixing multiplier

$b_L$ : relative transmissibility of individuals with latent infections

$\delta$ : rate of serological testing

$1/\kappa$ : average time to receive RNA test

$1/\xi$ : duration of test validity

**Figure 2.** Compartmental model for certifying infection status. (Online version in colour.)

standard *SEIR* model when $\alpha = 0$, $\kappa = 0$, $q = 0$, $b_{I_t} = 0$, $b_{L_u} = 0$ and $b_{L_t} = 0$. The parameters $q$, $\kappa$ and $\alpha$ are considered to be control parameters, while the remaining parameters are considered to be natural.

## (b) Strategy 2: Certification

The system of equations for Strategy 2 is

$$\dot{S}_c = \kappa S_u - \xi S_c - \beta(I_c + mI_u + b_L L_c + mb_L L_u)S_c, \tag{1.5a}$$

$$\dot{L}_c = \beta(I_c + mI_u + b_L L_c + mb_L L_u)S_c - \xi L_c - \sigma L_c, \tag{1.5b}$$

$$\dot{I}_c = \sigma L_c - \xi I_c - \gamma I_c, \tag{1.5c}$$

$$\dot{R}_c = \gamma I_c + \delta R_u, \tag{1.5d}$$

$$\dot{S}_u = \xi S_c - \kappa S_u - \beta(I_u + mI_c + b_L L_u + mb_L L_c)S_u, \tag{1.5e}$$

$$\dot{L}_u = \beta(I_u + mI_c + b_L L_u + mb_L L_c)S_u + \xi L_c - \sigma L_u - \kappa L_u, \tag{1.5f}$$

$$\dot{I}_u = \sigma L_u + \xi I_c - \gamma I_u - \kappa I_u \tag{1.5g}$$

$$\text{and} \quad \dot{R}_u = \gamma I_u + \kappa(I_u + L_u) - \delta R_u. \tag{1.5h}$$

This model (figure 2 and equation (1.5)) supposes that the overall population is subdivided into certified (subscript $c$) and uncertified (subscript $u$) subpopulations in which persons may be susceptible, incubating, symptomatic or removed/recovered (designated $S$, $L$, $I$ and $R$, respectively). As above, it is assumed that $L$ class individuals may contribute to the force of infection. The model allows that infection-free status may be conferred by either serological testing confirming past infection (durable certification) or having recently received a negative RNA test (temporary certification). We suppose that the primary purpose of certification is to change the patterns of contact between certified and uncertified people (i.e. with uncertified individuals practising intensive social distancing by sheltering in place, not going to school or work, and not participating in large gatherings). Furthermore, the role of RNA testing (at rate $\kappa$) differs between the two strategies. In Strategy 2, RNA testing is available to all uncertified individuals and is used to differentiate between susceptible ($S_u$) and infectious ($L_u$ and $I_u$) cases. Uncertified individuals receiving a positive RNA test are isolated from the population, represented by moving to the

uncertified removed compartment ($R_u$). All uncertified susceptible individuals have access to testing at the same rate.

In contrast to Strategy 1, here we assumed that the contact rate, $\beta$, is specific to within-class encounters (certified–certified or uncertified–uncertified) while infectious encounters between classes (certified-uncertified) occur at another rate ($\beta m$), where $m < 1$ is a factor that represents the reduction in mixing. Additionally, the factor $b_L < 1$ represents the reduced infectiousness of incubating infections compared with symptomatic infections. Therefore, the force of infection in the certified subpopulation is given by $\beta[I_c + b_L L_c + m(I_u + b_L L_u)]$ and for the uncertified population by $\beta[I_u + b_L L_u + m(I_c + b_L L_c)]$. There are two routes of certification. Temporary certification of susceptible individuals is achieved through an RNA test at rate $\kappa$. It is assumed that the temporary certification is valid for $1/\xi$ days. Durable certification through serological testing is conducted on individuals who have recovered from an infection at the rate $\delta$.

The parameters $\kappa$ and $\xi$ are considered to be control parameters, while the remaining parameters are considered to be natural.

Assuming that there is no certification process prior to the start of the epidemic (i.e. $S_c(0) = 0$, $S_u(0) = 1$ and $\kappa = 0$ initially), then the basic reproduction number for this model is

$$R_0 = \frac{\tilde{\beta} b_L}{\sigma} + \frac{\tilde{\beta}}{\gamma} \tag{1.6}$$

which is the standard form of the basic reproduction number for an SEIR-type compartmental model, where $\tilde{\beta} = \beta N$. If certification is initiated before the start of the epidemic (i.e. if $S_c(0) > 0$), we obtain a control reproduction number which, in general, is much more complicated. Let $R_{XY}$ denote the average number of new infections (in compartment $Y$) induced by the introduction of a single individual (in compartment $X$) into a completely susceptible population, over the course of the infectious period of this individual. Then the control reproduction number takes the form:

$$R_c = \frac{1}{2}(R_{L_c L_c} + R_{L_u L_u}) + \frac{1}{2}\sqrt{(R_{L_c L_c} - R_{L_u L_u})^2 + 4R_{L_c L_u} R_{L_u L_c}}. \tag{1.7}$$

Note that because infections always begin in the latent stage, new cases only appear in the latent compartments. Furthermore, the control reproduction number quantifies the average number of new cases induced by the introduction of a single 'average infectious individual' in a completely susceptible population. This average is determined by assuming the introduced infectious individual is equally likely to be latent or fully infectious and certified or uncertified. The details of this case and other observations can be found in electronic supplementary material, appendix S1.

The control reproduction number is strictly increasing with the inter-class mixing multiplier, $m$. If the classes do not mix at all ($m = 0$), then the control reproduction number is equal to the greater of the two direct reproduction numbers:

$$R_c = \max\{R_{L_c L_c}, R_{L_u L_u}\}$$
$$= \frac{\tilde{\beta}}{\sigma + \min\{\kappa, \xi\}} \left(\frac{\max\{\kappa, \xi\}}{\kappa + \xi}\right) \left(b_L + \frac{\sigma}{\gamma + \min\{\kappa, \xi\}}\right),$$

which is in general a lower bound for the value of the control reproduction number. On the other hand, if the certification process has no impact on the mixing between classes ($m = 1$), then the control reproduction number will be larger in general:

$R_c = R_{L_c L_c} + R_{L_u L_u}$. Similarly, this is an effective upper bound for $R_c$.

Looking at the other control parameters, the control reproduction number is strictly decreasing in the rate of certification testing, $\xi$. If certification remains valid indefinitely ($\xi = 0$), then it has no suppressive effect on overall transmission. In this case, the control reproduction number is equal to the basic reproduction number:

$$\lim_{\xi \to 0} R_c = R_0.$$

On the other hand, if certification has no effect ($\xi \to \infty$) then there is a significantly lower risk of an outbreak:

$$\lim_{\xi \to \infty} R_c = \frac{\tilde{\beta} b_L}{\sigma + \kappa} + \frac{\tilde{\beta}}{\gamma + \kappa} \left(\frac{\sigma}{\sigma + \kappa}\right).$$

Hence, without certification, the ability to prevent an outbreak is determined solely by the rate of RNA testing ($\kappa$).

We obtain similar results for the limiting cases for the rate of RNA testing ($\kappa$):

$$\lim_{\kappa \to 0} R_c = R_0$$

and

$$\lim_{\kappa \to \infty} R_c = \frac{\tilde{\beta} b_L}{\sigma + \xi} + \frac{\tilde{\beta}}{\gamma + \xi} \left(\frac{\sigma}{\sigma + \xi}\right).$$

## (c) Generalized interventions

Because transmission is the result of contagious contact, targeted and generalized interactions have multiplicative effects. In our model, generalized interventions are represented by multiplying $\beta$ by a factor less than one.

## (d) Implementation

Solutions to the equations were obtained using the R package `pomp` [30]. Much remains unknown about the epidemiology of COVID-19 in different settings and under various interventions and estimates of key parameters are therefore highly variable. In what follows, we parameterize our models according to what we believe are reasonable assumptions for typical settings in high income countries. All baseline parameters are shown in table 1 and explained here. We begin with the parameters we consider to be most well identified. We assume that the population comprises 10 million individuals ($N = 10^7$). Throughout, we assume the basic reproduction number, $R_0$, to be 2.5, consistent with numerous estimates [31] and the US CDC pandemic planning scenarios [20]. Following [32], we assume a presymptomatic incubation period of 5.1 days ($\sigma = 1/5.1$). The generation time, $G$, is equal to the sum of the incubation period and the symptomatic infectious period: $G = 1/\sigma + 1/\gamma$. Most estimates of the generation time (and the more commonly estimated serial interval) range from 5–7 days. We therefore assume $G = 6$ and rearrange to obtain $\gamma = 1/(G - 1/\sigma) = 1/(6 - 5.1) = 1/0.9$. We recognize that this is a much shorter infectious period than assumed by most studies, but believe it to be the assumption most consistent with the estimates in [31] and also in keeping with the evolving understanding of pre- versus post-symptomatic transmission [33]. We assume that incubating infections are 44% as contagious as symptomatic infections [34] so $b_{L_u} = 0.44$. Rearranging equation (1.6) to obtain

**Table 1.** Model parameters.

| parameter | value | meaning |
|---|---|---|
| $\beta$ | 0.8 | transmissibility |
| $b_{I_t}$ | 0.22 | factor to reduce transmissibility when traced |
| $b_{L_u}$ | 0.44 | factor to reduce transmissibility when latent |
| $b_{L_t}$ | 0.10 | factor to reduce transmissibility when latent and traced |
| $b_L$ | 0.44 | factor to reduce transmissibility when latent |
| $m$ | 0.22 | reduction in mixing due to certification |
| $\gamma$ | 1/0.9 | recovery rate |
| $\sigma$ | 1/5.1 | rate of progression through incubation period |
| $N$ | $1 \times 10^7$ | population size |
| $q$ | variable | case ascertainment |
| $\alpha$ | variable | contact tracing rate |
| $\kappa$ | variable | testing rate |
| $\xi$ | variable | testing expiration rate |
| $\delta$ | variable | rate of serological testing |

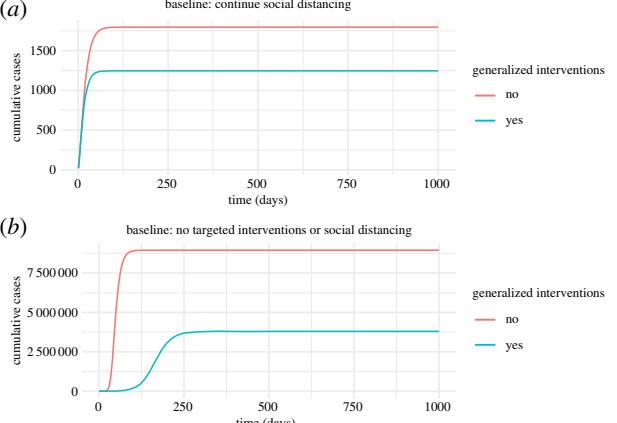

**Figure 3.** Two baseline scenarios which do not include targeted interventions. The top plot assumes that transmissibility, $\beta$, is at 22% of its original value due to social distancing. The bottom plot assumes that transmissibility, $\beta$, is at its natural value ($\beta N = 0.8$). Both plots assume that generalized interventions reduce transmissibility by a further 50%. Here, $q = 0$, $\kappa = 0$, $\delta = 1/10$, $\xi = 0$ and all other parameters are as given in table 1. Initially, there are 500 latently infected individuals, 500 recovered individuals, with the remainder susceptible. (Online version in colour.)

$\beta = R_0 \gamma \sigma / (b_{L_u} \gamma + \sigma) N \approx 8 \times 10^{-8}$. We assume that isolation eliminates non-household contacts. Specifically, we assumed the pre-isolation contact rate of 13.4 persons per day [35] was reduced to the global average household size of 4 (https://www.prb.org/international/indicator/hh-size-av/map/country) less 1 (for the patient in isolation) yielding $b_{I_t} = (4 - 1)/13.4 \approx 0.22$. Transmissibility reductions for incubation and isolation are assumed to be independent so $b_{L_t} = b_{L_u} \times b_{I_t} \approx 0.1$. For comparability between Strategy 1 and Strategy 2, we assume that the reduction in mixing due to certification is similar to the reduction in transmission due to isolation and set $m = b_{I_t} = 0.22$. Social distancing (i.e. 'lockdown') is assumed to reduce ($\beta$) by the same amount as isolation ($b_{I_t}$), which is also in agreement with estimates of [36] for Europe.

The effect of generalized interventions is difficult to estimate as these encompass a set of policies with varying levels of adherence and impact. In New York, face covering and mask wearing contributed to decreases in transmission ranging from 3.4 to 6.6%, while stay-at-home measures and school closures led to a decrease of up to 70% of transmission [37]. Another study found that voluntary self-isolation resulted in a reduction of transmission of 32% with household-level quarantine increasing the reduction to 37% [5]. Throughout the USA, bans on large gatherings, school closures and promotion of remote working led to a reduction of daily contacts between 65 and 75% [38]. Here, generalized interventions are assumed to reduce transmission by half (50%). Other modelling studies have used similar values: 25% and 50% [39], 25%, 50% and 75% [12].

Typically, we study the sensitivity of the final epidemic size to the choice of control parameters $q$, $\alpha$, $\kappa$, $\xi$ and $\delta$, but consider the values $q = 0.5$, $\alpha = 10 \times \beta = 8 \times 10^{-7}$, $\kappa = 1/3$, $\xi = 1/7$ and $\delta = 1/10$ as a reference point, implying case finding of 50%, that five contacts are traced for every secondary infection, that the delay to obtain a diagnostic test is 3 days, that diagnostic certification is valid for 7 days, and that the time to obtain an antibody test is 10 days. For comparison, we note that the CDC considers 50% to be the upper bound on

the percentage of cases that are asymptomatic [20,40]. The sensitivity of our conclusions to these choices is studied in greater detail in electronic supplementary material, appendix S2. We assume transmission is initiated with 1000 infected individuals evenly distributed between incubating and symptomatic compartments of the non-target class (i.e. untraced or uncertified).

## 3. Results

### (a) How much might generalized interventions (without targeted interventions) reduce the total outbreak size compared with reference scenarios?

The reference condition of continued social distancing (lockdown) is represented in the certification model by setting $\xi$ and $\kappa$ to 0 and setting $\beta_0$ to 22% of its original value. (For comparison, equivalent baseline conditions for the contact tracing model are provided in electronic supplementary material, appendix S2, figures S3, S6 and S10.) At the assumed level of social distancing, an outbreak still occurs, ultimately infecting a little under 2% of the population (figure 3, top). Social distancing combined with generalized interventions does not result in complete suppression, but reduces transmission to very close to the critical level.

A scenario with no social distancing and no targeted interventions is represented by setting $\xi$ and $\kappa$ to 0 and $m = 1$ (electronic supplementary material, figure S1). Unsurprisingly, the large majority of the population is infected under this condition. Generalized interventions (i.e. mask wearing, improved hand hygiene, behavioural changes) reduce the total outbreak size by about 58% (figure 3, bottom). These results suggest that generalized interventions of the magnitude envisioned here are not sufficient to suppress transmission to the same extent as continued social distancing. If continued social distancing is not possible, then targeted interventions will be essential if infection of the majority of the population is to be prevented.

## (b) When are contact tracing and quarantine most beneficial?

Active case finding, contact tracing and quarantine represent an escalation of Strategy 1 approaches to suppressing transmission. When untraced individuals are identified through active case finding or contact tracing, they are then traced and isolated, reducing their ability to transmit. On the other hand, traced infectious individuals in quarantine do not contribute at all to transmission. As a baseline, it is therefore useful to understand the conditions, if any, under which active case finding alone can limit transmission. To investigate active case finding as a control parameter, we set $\alpha = 0$ and $\kappa = 0$ and plot the final epidemic size as a function of $q$. Case finding cannot lead to complete suppression without generalized interventions (figure 4, red line). The addition of generalized interventions provides a critical value for case finding of around 90% (figure 4, all other lines), which seems untenable for a disease that is symptomatic in only around 80% of cases (the effect of active case finding with no contact tracing is illustrated in electronic supplementary material, appendix S2 and figure S5). At a more realistic level of 50% case finding, greater than 20% of the population would be infected with generalized interventions and around 85% without generalized interventions. For comparison, many scientists and health experts think case ascertainment of COVID-19 in a number of settings was originally between 1% and 10% [41,42], so 50% represents finding about five times as many cases as occurred during the first wave. It seems implausible that 50% case finding could occur without widespread testing. We also show the relative impact of contact tracing and quarantine (figure 4, blue and purple lines). For parameters studied here, the relative additional benefits provided by quarantine or contact tracing are quite small compared with generalized interventions and active case finding. Further, contact tracing and quarantine do not change the value of case finding at which suppression is achieved, but do reduce the total number of cases for a given level of case finding below the critical value of $q^*$.

## (c) What benefit does quarantine add to contact tracing?

These results are possibly surprising. Particularly, why is quarantine not more effective compared with contact tracing, given that it has been such a long-standing public health strategy? Our model assumes that quarantined individuals are excluded from encounters in the general population. But, in recognition that traced contacts will often be family members and expecting that family members may be quarantined together, the model allows for transmission at 10% of the baseline value. We wondered if this small amount of transmission from quarantined individuals to family members accounts for the difference. To investigate this idea, we repeated the analysis setting $b_{I_t} = 0$ and $b_{L_t} = 0$, turning off transmission to or from traced contacts entirely. The overall shape of the effect of case identification on total outbreak size is similar, but shifted (figure 5).

Comparing the curves in figures 4 and 5 provides an idea of the relative impact of perfect isolation on the efficacy of each strategy. Eliminating transmission from traced groups increases the total number of cases averted approximately 10-fold from 250 000 to almost 1 000 000, as the fraction of

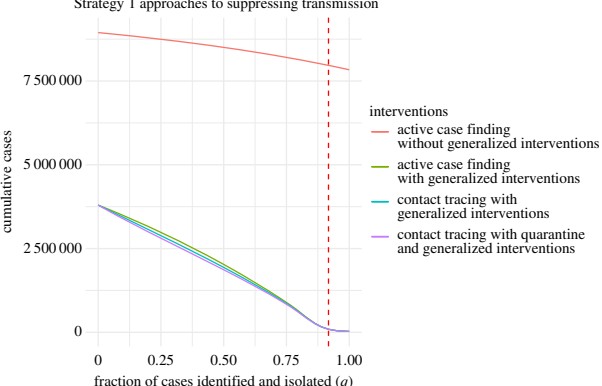

**Figure 4.** Strategy 1 approaches to suppressing COVID-19 transmission as a function of case ascertainment ($q$). The red dashed line shows the critical value $q^*$ at which suppression is achieved ($R_c = 1$) with generalized interventions. Generalized interventions reduce transmissibility by a further 50%. Without generalized interventions, suppression by active case finding is not possible. Approaches with contact tracing take $\alpha = 10 \times \beta$ and $\kappa = 1/3$. The quarantine scenario assumes that $b_{I_t} = 0$ and $b_{L_t} = 0$. Other parameters are as in table 1. (Online version in colour.)

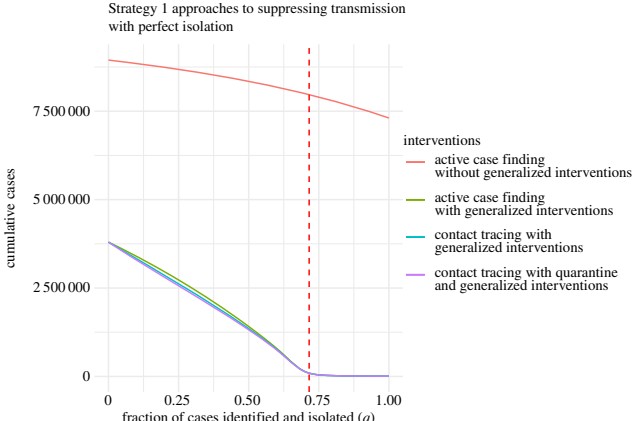

**Figure 5.** Strategy 1 approaches to suppressing COVID-19 transmission with perfect isolation ($b_{L_t} = 0$ and $b_{I_t} = 0$) as a function of case ascertainment ($q$). The red dashed line shows the critical value $q^*$ at which suppression is achieved ($R_c = 1$) with generalized interventions. Generalized interventions reduce transmissibility by a further 50%. Without generalized interventions, suppression by active case finding is not possible. Approaches with contact tracing take $\alpha = 10 \times \beta$ and $\kappa = 1/3$. The quarantine scenario assumes that $b_{I_t} = 0$ and $b_{L_t} = 0$. Other parameters are as in table 1. (Online version in colour.)

cases identified and isolated ($q$) varies from 25 to 75% for all three Strategy 1 approaches (figure 6). The steep drop-off of cases averted is due to the fact that cumulative cases are significantly smaller when the fraction of cases identified is very large.

In electronic supplementary material, appendix S2, we also illustrate the relationship between outbreak size, tracing rate and testing intensity with and without generalized interventions and quarantine (electronic supplementary material, figures S8, S9, S12 and S13).

## (d) When can certification be effective?

Here we look at certification with and without generalized interventions. The baseline number of cumulative cases are

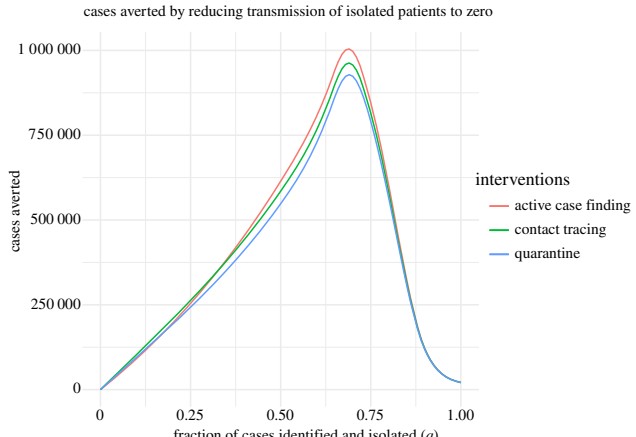

**Figure 6.** Cases averted by reducing transmission from isolated patients from 10% to zero ($b_{L_t} = 0$, $b_{l_t} = 0$) as a function of case ascertainment ($q$). Other parameters are as described in figure 4. (Online version in colour.)

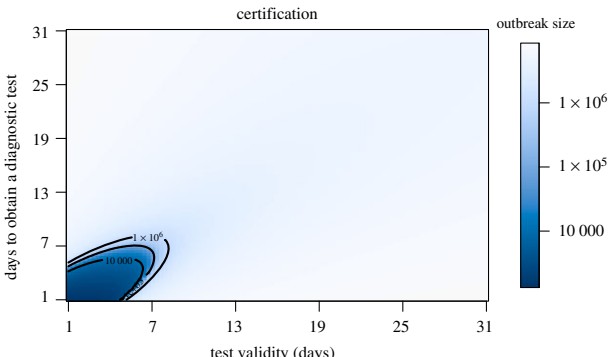

**Figure 7.** Final outbreak size as a function of viral test validity duration ($1/\xi$) and test waiting time ($1/\kappa$) without generalized interventions. At the assumed value of presymptomatic transmission ($b_L = 0.44$), there is only a very small region (dark blue) within which certification can prevent a major epidemic. Outbreak size is given on a $\log_{10}$ scale. Other parameters are as in table 1. (Online version in colour.)

displayed in electronic supplementary material, appendix S2 and figure S14. To better understand the range of conditions under which certification can be effective, we examine the final outbreak size over a grid comprising all combinations of viral test validity from 1 to 14 days and for test waiting times from 1 to 14 days (figure 7). Interestingly, in both cases there is a very sharp boundary between those testing regimes in which suppression of transmission is achieved (dark blue) and testing regimes where a very large outbreak ensues. Generally, test validity duration and test waiting times must be shorter than a week to suppress transmission. These results suggest that it is virtually impossible to suppress transmission without generalized interventions. The 'safe' region (dark blue) is substantially larger when there are generalized interventions (figure 8). Specifically, it appears that a test validity of up to 24 days together with a waiting time of no more than 19 days would achieve suppression. However, the sharpness of the boundary between suppression and a failure to suppress suggests that this approach is fragile, such that small inaccuracies in parameter values or model specification may cause the approach to fail. Furthermore, this result relies on a substantial reduction in mixing (of 78%) between certified and non-certified populations.

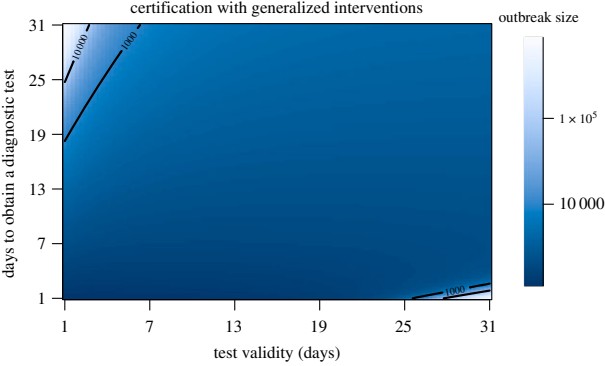

**Figure 8.** Final outbreak size as a function of viral test validity duration ($1/\xi$) and test waiting time ($1/\kappa$) with generalized interventions. At the assumed value of presymptomatic transmission ($b_L = 0.44$), there is only a very small region (dark blue) within which certification can prevent a major epidemic. Outbreak size is given on a $\log_{10}$ scale. Other parameters are as in table 1. (Online version in colour.)

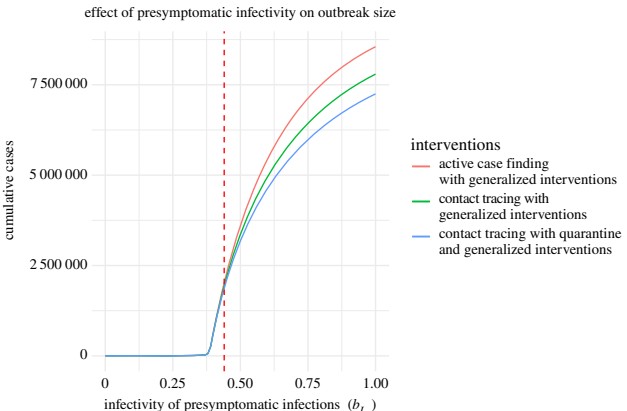

**Figure 9.** Effect of presymptomatic infectivity on outbreak size for $0 < b_{L_u} < 1$. The vertical dashed line shows the default value of $b_{L_u} = 0.44$ for comparison with other figures. Generalized interventions are assumed to reduce $\beta$ by 50%. Other parameters are as in table 1 and $q = 0.5$. (Online version in colour.)

### (e) How does the extent of presymptomatic transmission affect the choice of intervention strategy?

The preceding analyses assume that latent cases are 44% as infectious as symptomatic cases, but it is well known that 'silent transmission' is a key component of COVID-19 epidemiology [24,25]. Here we investigate how different levels of presymptomatic transmission influence the effectiveness of the containment strategies introduced here. First, we plot the total outbreak size against the assumed level of infectivity (i.e. the parameter $b_{L_u}$); for each level of $b_{L_u}$, the transmissibility of traced individuals is set to $b_{L_t} = b_{I_t} \times b_{L_u}$ (figure 9).

Next we look at the certification model at four different levels of $b_L$. Epidemic outcomes are summarized by plotting the contour for combinations of test validity ($1/\xi$) and test lag ($1/\kappa$) where the final outbreak size is 10 000 (figure 10). Regions bounded by a contour line represent combinations of test lag and validity which can suppress transmission for the given value of $b_L$. Because the transition is so sharp (compare figure 8), this is effectively the 'containment boundary' separating minor transmission and a major epidemic. Unsurprisingly, for presymptomatic transmission less than the default value of 0.44, a longer test validity and test lag may

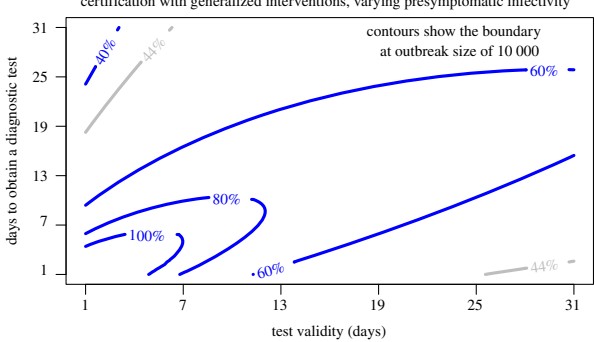

**Figure 10.** Certification with generalized interventions for presymptomatic infectivity ($b_L$) assumed to be 40%, 44%, 60%, 80% and 100% of the baseline value of $\beta \times N = 0.8$ for transmission with generalized interventions. Combinations of test validity and days to obtain a test that are below the contour have total outbreak sizes less than 10 000 and may be considered to be 'contained'. The default value of $b_L = 0.44$ is plotted in grey for comparison with figure 8. Other parameters are $m = 0.1$, $\gamma = 1/6$, $\sigma = 1/4$ and $\delta = 0.1$. (Online version in colour.)

be tolerated without risking a major outbreak (these regions fall outside the range of the figure). However, even with no presymptomatic transmission (0% contour), the safe region remains relatively small with a maximum test validity of around two weeks. As presymptomatic transmission approaches the level of symptomatic transmission, the safe region diminishes substantially.

## 4. Discussion

While we await the distribution and further development of vaccines and other pharmaceutical interventions, and probably even after their arrival, non-pharmaceutical approaches to minimizing the impact of COVID-19 will remain important. The detection of a variant of SARS-CoV-2 with increased transmissibility further highlights the need to explore the role of NPIs in suppressing transmission [43]. Due to their current and ongoing importance, NPIs have received considerable attention and have been modelled extensively. Most previous studies have focused on the impact that population-wide, generalized interventions, particularly reduced mobility though 'lockdowns', have had on outbreak dynamics [3,7,8,12,36, 44,45]. Recently, the impact of targeted interventions, namely testing, contact tracing and household quarantine, were explored for the Boston area, and it was found that strong targeted interventions might keep the pathogen under control [9]. Similar studies, focused on the UK, also found that isolation and contact tracing might allow control of the pathogen if deployed with sufficient intensity [2,5]. Finally, at a smaller scale, a few models have been developed to understand how to limit transmission in the setting of university campuses. In these, screening (the testing of community members before admittance to campus) has been added to the targeted interventions of regular testing and contact tracing [46,47]. In all of these studies, containment required achieving very high levels of detection.

While we are not aware of other studies that consider temporary certification, ideas analogous to our concept of durable certification have been studied. The idea is that individuals certified to have recovered from infection (and therefore presumed to be immune) could be placed into positions that substitute their contacts with those of non-certified individuals. This has been recently explored under the concept of 'shield immunity' [16] or 'cocooning' of vulnerable populations [10].

It is clear from dramatic declines in cases in places that have implemented sustained lockdowns that eliminating contact to the most feasible extent can reduce the spread of SARS-CoV-2 [7,36,48]. Anecdotal evidence exists for the effect of other generalized interventions, such as the use of face masks [49], although quantitative estimates for their effectiveness at the population level currently seem to not exist. The relative impact of targeted interventions such as screening or testing programmes is much harder to quantify and to separate from generalized interventions. Our study suggests that their effects may be quite substantial.

### (a) Are our parameters realistic for COVID-19?

In most of the scenarios studied, we assumed that active case finding would yield case ascertainment rates of 50%. For context, this can be compared with either estimated case ascertainment rates or estimated symptomatic rates (which sets an upper bound on case acertainment through clinical diagnosis). In analysis of data from passengers on the *Diamond Princess* cruise ship, Mizumoto *et al.* [50] estimated an overall asymptomatic proportion of 17.9% (equating to a symptomatic proportion of 82.1%). Among residents in a nursing home, 10 out of 23 (43.5%) were symptomatic at the time of testing [51]. A review of multiple populations found that the fraction of asymptomatic persons infected with SARS-CoV-2 may be 45–50% [40]. For comparison, estimates of ascertainment in the USA in for spring 2020 are in the range of 1–10% [41,42].

### (b) Strategic approaches to suppressing transmission

Here we have analysed two structurally different approaches to suppressing transmission without intensive social distancing (i.e. 'lockdowns'). Without employing either of these approaches, generalized interventions will reduce cumulative cases but these approaches are much more effective if social distancing can be maintained over a long period. The ability for Strategy 1 approaches (contact tracing and quarantine) to decrease the proportion of the population infected is driven primarily by the level of case finding. These approaches are much more effective when generalized interventions are implemented. However, Strategy 1 approaches are not likely to fully suppress of transmission: this would require case finding to detect approximately 90% of cases even with generalized interventions. The addition of perfect quarantine to Strategy 1, where traced contacts are fully separated from the rest of the population, leads to a considerable increase of the number of cases averted. The Strategy 2 approach to employ a certification process can be effective at reducing outbreak sizes in two scenarios. Without generalized interventions, the duration of test validity and the test waiting time must both be less than one week to achieve suppression. Unsurprisingly, the certification process is much more effective at suppression if generalized interventions are also employed. Outbreak size can be decreased 10-fold with test validity durations and waiting times less than a month in this case (figure 8).

The effectiveness of both strategies varies significantly with the assumed level of presymptomatic infectivity. In the case of Strategy 1, higher levels of presymptomatic infectivity

increases the relative benefit of quarantine. For Strategy 2, increased presymptomatic infectivity decreases the range of effective test validity and waiting times. For example, with generalized interventions, a presymptomatic infectivity of 80% would require these times to be under two weeks compared with under a month in the baseline case.

The two strategies are distinct because their flow diagrams (figures 1 and 2) are incommensurate and neither is a special or limiting case of the other. Of course, these strategies could be used together for greater effectiveness. However, a flow model to evaluate the optimal use of strategies in combination would be considerably more complicated, requiring approximately 16 states to represent the possible combinations of certified and uncertified persons that may be either traced or untraced and in one of the four primary infection states ($S$, $L$, $I$ and $R$). Even though the actual number of flows among these 16 states will be considerably fewer than the $16 \times 15 = 240$ possibilities, it would be a considerable challenge to sensibly parametrize such a model. Developing such a model could nonetheless be a useful future step toward developing a complete understanding of transmission reduction via non-pharmaceutical interventions for acute infectious diseases.

## (c) Conclusions

These results suggest that any of the preceding strategies may suppress transmission, but that suppression depends on achieving a certain level of effectiveness (reduction in transmission among isolated persons, intensity of contact tracing, frequency of certification, etc.) that varies according to the strategy. Particularly, Strategy 1 approaches (active case finding, contact tracing and quarantine) are expected to work only when case ascertainment is high. Because actual case finding rates are more in the range of 1–10% [41,42], this approach is unlikely to achieve the goal of full suppression unless our ability to locate cases markedly improves. We find that the value of quarantine relative to active case finding and contact tracing is surprisingly small. That said, this common and longstanding public health strategy does effectively increase the number of cases averted if it

fully isolates infected patients. The success of a Strategy 1 approach would thus require significant investment in improvements to case finding and contact tracing measures as well as enactment of policies for effective enforcement of quarantine. Similarly, Strategy 2 approaches (certification) are only expected to succeed in a narrow range of conditions (i.e. high frequency testing with low durations of certification validity). This approach may become feasible as ongoing efforts to increase COVID-19 test turnaround time proceed. However, scaling up to enable frequent testing of large populations would require a serious investment in both testing technology and epidemiological infrastructure. To our knowledge, this type of certification process has not been employed at large scale, though similar processes are used at smaller organizational levels. Studies may be conducted to ascertain the efficacy of these similar but small-scale processes to improve predictions of the success of a certification policy at population scale. These findings suggest that, regardless of whether Strategy 1 approaches or Strategy 2 approaches are adopted, a large testing capacity is required. Furthermore, success depends on the effectiveness of generalized interventions because, in the realistic scenarios that we consider, these will be essential to achieve suppression.

Data accessibility. This article has no additional data.

Authors' contributions. J.M.D. conceived the study, wrote the code and drafted the manuscript. P.R. contributed to model formulation, interpretation of results and derivation of $R_0$. K.D. contributed to the derivation of $R_0$. A.H. contributed to model formulation and interpretation of results. All authors contributed to editing the manuscript and gave final approval for publication and agree to be held accountable for the work performed therein.

Competing interests. We declare we have no competing interests.

Funding. Research reported here was supported by the National Institute of General Medical Sciences of the National Institutes of Health under award no. U01GM110744. A.H. was partially supported by NIH grant GM 12480-03S1. The content is solely the responsibility of the authors and does not necessarily reflect the official views of the National Institutes of Health.

Acknowledgements. We thank Eric Marty for assistance with figures 1 and 2, and Paige Miller for comments on an earlier version of this paper. Andrew Tredennick assisted with troubleshooting code.

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
