## [Peer Review File · Proceedings of the Royal Society B: Biological Sciences]

Review History

RSPB-2020-1839.R0 (Original submission)

Review form: Reviewer 1

Recommendation

Major revision is needed (please make suggestions in comments)

Scientific importance: Is the manuscript an original and important contribution to its field?

Good

General interest: Is the paper of sufficient general interest?

Good

Quality of the paper: Is the overall quality of the paper suitable?

Acceptable

Is the length of the paper justified?

Yes

Should the paper be seen by a specialist statistical reviewer?

Yes

Do you have any concerns about statistical analyses in this paper? If so, please specify them explicitly in your report.

No

It is a condition of publication that authors make their supporting data, code and materials available - either as supplementary material or hosted in an external repository. Please rate, if applicable, the supporting data on the following criteria.

Is it accessible?

N/A

Is it clear?

N/A

Is it adequate?

N/A

Do you have any ethical concerns with this paper?

No

Comments to the Author

In my opinion, this study is well conducted and its findings are interesting, since they compare five approaches to suppress the SARS-CoV-2 epidemic without intensive social distancing.

Different models and equations used in this study are well described, however their descriptions are not always easy to follow. I would suggest the Authors to improve the paragraphs related to the description of models. This could be useful, especially for readers without a statistical background.

Moreover, I would suggest to include appropriate references and descriptions of assumed parameters that regulate each model (e.g. transmission parameters, incubating period etc). Finally, I would ask if the authors considered the proportion of undocumented cases (please consider the following articles DOI: 10.3390/jcm9051350; DOI: 10.1126/science.abb3221). It is possible that different approach might be differentially affected by undocumented cases?

Review form: Reviewer 2

Recommendation

Reject – article is not of sufficient interest (we will consider a transfer to another journal)

Scientific importance: Is the manuscript an original and important contribution to its field?

Marginal

General interest: Is the paper of sufficient general interest?

Marginal

Quality of the paper: Is the overall quality of the paper suitable?

Marginal

Is the length of the paper justified?

Yes

Should the paper be seen by a specialist statistical reviewer?

No

Do you have any concerns about statistical analyses in this paper? If so, please specify them explicitly in your report.

No

It is a condition of publication that authors make their supporting data, code and materials available - either as supplementary material or hosted in an external repository. Please rate, if applicable, the supporting data on the following criteria.

Is it accessible?

No

Is it clear?

N/A

Is it adequate?

N/A

Do you have any ethical concerns with this paper?

No

Comments to the Author

This paper uses models to compare five distinct non-pharmaceutical strategies for controlling the COVID-19 pandemic without extreme social distancing (lockdowns). This is an interesting, important, and timely topic. This paper attempts to fill a somewhat narrow niche by using a general model based on SARS-CoV-2, without any specific parameter information, to inform general control strategies, but presenting results that mainly focus on quantitative (or semi-quantitative “ballpark”) results. The general results (e.g., the first sentence of the Conclusions) are predictable without the model. The general conclusions about the need for high rates of case ascertainment and narrow ranges of testing speed and durability are also self-evident. The main compelling results rest on semi-quantitative outcomes – for example, that 95% of cases must be ascertained to suppress the epidemic without generalized interventions, and that test results must come back within 3 days and last less than 2 weeks to be effective. But given that the paper does not discuss parameter estimation or couch the parameter values in published or publicly available sources, we can’t put much stock in these more quantitative results (even though it is tempting to do so). As written the Introduction, Discussion, and Conclusions provide insufficient background and context to really inspire broad interest.

The Introduction is very brief, and does not include an overview of previous modeling efforts that have addressed similar questions (of which there are many). Though it is difficult for anyone to stay on top of the COVID-19 modeling literature (especially as so much of it currently exists in preprints), it is important for this paper to identify a clear gap that it seeks to fill that other papers have not already considered, and to discuss the rationale for using a generic model not parameterized for any specific location to make inferences about the efficacy of different control strategies. For example, have any previous models evaluated some but not all of these strategies, or considered them individually but not together?

The equations presented in (1) are hard to follow. Why does the RNA testing rate affect only the transition from S_t to S_u and not from L_t to I_t ? Why do L_t individuals transition into L_u ? More build-up of the model approach would be useful here, going step by step, numbering and referring to the equations individually, and defining all parameters in the text and not just in Fig. 1.

Before diving into the analysis strategy (calculating R_0), the Methods should give an overview of the approach and goals of the models and analysis. Why is R_0 calculated here? Is that the metric of control that the analyses will use? This does not seem well aligned with the goal of studying the dynamics of the epidemic once it has emerged (especially for interventions that can affect the recovered class, such as certification). Or will simulations of dynamics also be used?

In the paragraph describing the interpretation of force of infection functions (page 5), referring to parameters by their biological interpretations and not just by symbols would make this section much easier to interpret. Giving the intuition for why the functions are interpreted as they are is necessary to allow a reader to follow the rationale.

In Model 2, why are certified individuals only assumed to enter the S class and not the R class, even though the text discusses “durable” certification based on antibody testing? It doesn’t seem to make sense to consider the effects of certification on R_0 , since a primary goal of (durable) certification is to identify individuals who are immune and can safely interact with others without risk of transmission. This focus on R_0 (page 7) leads to the somewhat misleading statement “if certification remains valid indefinitely ... then the risk of an outbreak is the same as if there were no certification process”, even though the focus of this paper is not on the probability of an outbreak but on the dynamics of that outbreak once it occurs. The statement “if certification has no effect ... then there is significantly lower risk of an outbreak” also does not really make sense; it’s not clear what ‘lower’ is relative to, or how the rate of RNA testing is assumed to affect transmission (via isolation of infected people, presumably, but this isn’t spelled out in the description of the model).

The “Methods: Generalized interventions” sub-section needs more information. Which intervention levels are considered, and why? How does this compare to estimated reductions in R_{eff} that have occurred since the start of the pandemic? Determining reasonable levels of these interventions is the key element to determining the relative effectiveness of these different intervention strategies.

Under the Implementation section, why was pomp used? Is aspect of the model stochastic? Were there any parameters estimated, from any data? Again, we need better context about what purpose the analyses served and what their main approaches were, before diving into the details. How were the parameter values chosen? There is plenty of parameter information publicly available (e.g., through the MIDAS network COVID-19 modeling portal), so further justification of parameter choices is needed. Without knowing how realistic the parameter values are, it is unclear how this model can inform the COVID-19 response.

In the second paragraph of the Results section, what are the “generalized interventions” used here? What does this mean in terms of parameter values and in terms of interpretation (e.g., mask-wearing, physical distancing, etc.)? Since generalized interventions are important throughout most of the results, it is important to revisit what is assumed here.

In the Results, the choice of parameter values becomes important in making statements like the last sentence on page 8 (95% of cases have to be identified to achieve suppression). This is number is likely to be sensitive to the choice of transmission rates, recovery rates, etc. It would also be helpful (in addition to Fig. 4) to see simulated dynamics for these intervention scenarios (as incident cases, rather than cumulative cases), so we can see how interventions affect the epidemic peak and how long they need to be maintained for the epidemic to burn out completely (if ever).

On page 10, what is the distinction between contact tracing and quarantine? This is worth restating here.

For the “When can certification be effective?” results, how dependent is this on the proportion of people who can receive testing (undergo possible certification)? Is there a tradeoff between

proportional testing coverage and testing validity and wait times?

The Discussion section is insufficient. It needs a general overview of the key results and their implications for COVID-19 control. It needs to place the results in context with other COVID-19 literature. It needs to discuss possible model sensitivity to parameters and other assumptions. For example, the section "Are our parameters realistic for COVID-19?" only discusses the asymptomatic fraction, and does not discuss any other important parameters that are likely to affect the results, such as transmission rates, recovery rates, mixing rates, and relative infectiousness rates. It needs to discuss how different intervention types have been applied already, and how successful they have been.

Minor comments:

Introduction and Discussion: "represented by incommensurable flow diagrams" is an opaque way of phrasing this.

In the Model 2 methods on page 5, the first sentence following equations (4) is unclear (both certified and uncertified persons who are S, L, and I, but two pools of R).

The distinction between Figs. 4 and 5 should be made clearer in the caption.

Page 11 "by comparing the curves in Figures 4 and 5" is not a very effective way to present results, given that the two figures look almost identical but with different y-axis scaling. Isn't the difference between them what's plotted in Fig. 6, a much better visualization of this result?

The Appendices need to be written out with equation numbers and figure numbers, as in the main text, and these figures need to be referred to in the main text. There is not enough context in Appendix 2 to interpret these figures.

Decision letter (RSPB-2020-1839.R0)

17-Aug-2020

Dear Professor Drake,

We have now received referees' reports on your manuscript RSPB-2020-1839 entitled "Five approaches to the suppression of SARS-CoV-2 without intensive social distancing".

As you will see, the referees differ in their evaluation, and the more negative reviewer has raised a number of concerns with the paper and indicated that substantial revisions are necessary. The manuscript has therefore been rejected in its current form. We would be happy to consider a resubmission, provided the comments of the referees are fully addressed. I fully appreciate the timeliness and importance of the work, but the reviewer has raised some major criticisms of the manuscript. Therefore please note that this is not a provisional acceptance.

Finally, I hope you and your co-authors are well in these challenging times.

Yours sincerely,
 Professor Loeske Kruuk
 Editor
 mailto: proceedingsb@royalsociety.org

Reviewer(s)' Comments to Author:

Referee: 1

Comments to the Author(s)

In my opinion, this study is well conducted and its findings are interesting, since they compare five approaches to suppress the SARS-CoV-2 epidemic without intensive social distancing. Different models and equations used in this study are well described, however their descriptions are not always easy to follow. I would suggest the Authors to improve the paragraphs related to the description of models. This could be useful, especially for readers without a statistical background.

Moreover, I would suggest to include appropriate references and descriptions of assumed parameters that regulate each model (e.g. transmission parameters, incubating period etc). Finally, I would ask if the authors considered the proportion of undocumented cases (please consider the following articles DOI: 10.3390/jcm9051350; DOI: 10.1126/science.abb3221). It is possible that different approach might be differentially affected by undocumented cases?

Referee: 2

Comments to the Author(s)

This paper uses models to compare five distinct non-pharmaceutical strategies for controlling the COVID-19 pandemic without extreme social distancing (lockdowns). This is an interesting, important, and timely topic. This paper attempts to fill a somewhat narrow niche by using a general model based on SARS-CoV-2, without any specific parameter information, to inform general control strategies, but presenting results that mainly focus on quantitative (or semi-quantitative "ballpark") results. The general results (e.g., the first sentence of the Conclusions) are predictable without the model. The general conclusions about the need for high rates of case ascertainment and narrow ranges of testing speed and durability are also self-evident. The main compelling results rest on semi-quantitative outcomes—for example, that 95% of cases must be ascertained to suppress the epidemic without generalized interventions, and that test results must come back within 3 days and last less than 2 weeks to be effective. But given that the paper does not discuss parameter estimation or couch the parameter values in published or publicly available sources, we can't put much stock in these more quantitative results (even though it is

tempting to do so). As written the Introduction, Discussion, and Conclusions provide insufficient background and context to really inspire broad interest.

The Introduction is very brief, and does not include an overview of previous modeling efforts that have addressed similar questions (of which there are many). Though it is difficult for anyone to stay on top of the COVID-19 modeling literature (especially as so much of it currently exists in preprints), it is important for this paper to identify a clear gap that it seeks to fill that other papers have not already considered, and to discuss the rationale for using a generic model not parameterized for any specific location to make inferences about the efficacy of different control strategies. For example, have any previous models evaluated some but not all of these strategies, or considered them individually but not together?

The equations presented in (1) are hard to follow. Why does the RNA testing rate affect only the transition from S_t to S_u and not from L_t to L_u ? Why do L_t individuals transition into L_u ? More build-up of the model approach would be useful here, going step by step, numbering and referring to the equations individually, and defining all parameters in the text and not just in Fig. 1.

Before diving into the analysis strategy (calculating R_0), the Methods should give an overview of the approach and goals of the models and analysis. Why is R_0 calculated here? Is that the metric of control that the analyses will use? This does not seem well aligned with the goal of studying the dynamics of the epidemic once it has emerged (especially for interventions that can affect the recovered class, such as certification). Or will simulations of dynamics also be used?

In the paragraph describing the interpretation of force of infection functions (page 5), referring to parameters by their biological interpretations and not just by symbols would make this section much easier to interpret. Giving the intuition for why the functions are interpreted as they are is necessary to allow a reader to follow the rationale.

In Model 2, why are certified individuals only assumed to enter the S class and not the R class, even though the text discusses “durable” certification based on antibody testing? It doesn’t seem to make sense to consider the effects of certification on R_0 , since a primary goal of (durable) certification is to identify individuals who are immune and can safely interact with others without risk of transmission. This focus on R_0 (page 7) leads to the somewhat misleading statement “if certification remains valid indefinitely ... then the risk of an outbreak is the same as if there were no certification process”, even though the focus of this paper is not on the probability of an outbreak but on the dynamics of that outbreak once it occurs. The statement “if certification has no effect ... then there is significantly lower risk of an outbreak” also does not really make sense; it’s not clear what ‘lower’ is relative to, or how the rate of RNA testing is assumed to affect transmission (via isolation of infected people, presumably, but this isn’t spelled out in the description of the model).

The “Methods: Generalized interventions” sub-section needs more information. Which intervention levels are considered, and why? How does this compare to estimated reductions in R_{eff} that have occurred since the start of the pandemic? Determining reasonable levels of these interventions is the key element to determining the relative effectiveness of these different intervention strategies.

Under the Implementation section, why was pomp used? Is aspect of the model stochastic? Were there any parameters estimated, from any data? Again, we need better context about what purpose the analyses served and what their main approaches were, before diving into the details. How were the parameter values chosen? There is plenty of parameter information publicly available (e.g., through the MIDAS network COVID-19 modeling portal), so further justification of parameter choices is needed. Without knowing how realistic the parameter values are, it is unclear how this model can inform the COVID-19 response.

In the second paragraph of the Results section, what are the “generalized interventions” used here? What does this mean in terms of parameter values and in terms of interpretation (e.g., mask-wearing, physical distancing, etc.)? Since generalized interventions are important throughout most of the results, it is important to revisit what is assumed here.

In the Results, the choice of parameter values becomes important in making statements like the last sentence on page 8 (95% of cases have to be identified to achieve suppression). This number is likely to be sensitive to the choice of transmission rates, recovery rates, etc. It would also be helpful (in addition to Fig. 4) to see simulated dynamics for these intervention scenarios (as incident cases, rather than cumulative cases), so we can see how interventions affect the epidemic peak and how long they need to be maintained for the epidemic to burn out completely (if ever).

On page 10, what is the distinction between contact tracing and quarantine? This is worth restating here.

For the “When can certification be effective?” results, how dependent is this on the proportion of people who can receive testing (undergo possible certification)? Is there a tradeoff between proportional testing coverage and testing validity and wait times?

The Discussion section is insufficient. It needs a general overview of the key results and their implications for COVID-19 control. It needs to place the results in context with other COVID-19 literature. It needs to discuss possible model sensitivity to parameters and other assumptions. For example, the section “Are our parameters realistic for COVID-19?” only discusses the asymptomatic fraction, and does not discuss any other important parameters that are likely to affect the results, such as transmission rates, recovery rates, mixing rates, and relative infectiousness rates. It needs to discuss how different intervention types have been applied already, and how successful they have been.

Minor comments:

Introduction and Discussion: “represented by incommensurable flow diagrams” is an opaque way of phrasing this.

In the Model 2 methods on page 5, the first sentence following equations (4) is unclear (both certified and uncertified persons who are S, L, and I, but two pools of R).

The distinction between Figs. 4 and 5 should be made clearer in the caption.

Page 11 “by comparing the curves in Figures 4 and 5” is not a very effective way to present results, given that the two figures look almost identical but with different y-axis scaling. Isn’t the difference between them what’s plotted in Fig. 6, a much better visualization of this result?

The Appendices need to be written out with equation numbers and figure numbers, as in the main text, and these figures need to be referred to in the main text. There is not enough context in Appendix 2 to interpret these figures.

Author's Response to Decision Letter for (RSPB-2020-1839.R0)

See Appendix A.

RSPB-2020-3074.R0

Review form: Reviewer 1 (Andrea Maugeri)

Recommendation

Accept as is

Scientific importance: Is the manuscript an original and important contribution to its field?

Good

General interest: Is the paper of sufficient general interest?

Good

Quality of the paper: Is the overall quality of the paper suitable?

Good

Is the length of the paper justified?

Yes

Should the paper be seen by a specialist statistical reviewer?

No

Do you have any concerns about statistical analyses in this paper? If so, please specify them explicitly in your report.

No

It is a condition of publication that authors make their supporting data, code and materials available - either as supplementary material or hosted in an external repository. Please rate, if applicable, the supporting data on the following criteria.

Is it accessible?

N/A

Is it clear?

Yes

Is it adequate?

Yes

Do you have any ethical concerns with this paper?

No

Comments to the Author

None

Review form: Reviewer 2

Recommendation

Accept with minor revision (please list in comments)

Scientific importance: Is the manuscript an original and important contribution to its field?

Excellent

General interest: Is the paper of sufficient general interest?

Excellent

Quality of the paper: Is the overall quality of the paper suitable?

Excellent

Is the length of the paper justified?

Yes

Should the paper be seen by a specialist statistical reviewer?

No

Do you have any concerns about statistical analyses in this paper? If so, please specify them explicitly in your report.

No

It is a condition of publication that authors make their supporting data, code and materials available - either as supplementary material or hosted in an external repository. Please rate, if applicable, the supporting data on the following criteria.

Is it accessible?

Yes

Is it clear?

Yes

Is it adequate?

Yes

Do you have any ethical concerns with this paper?

No

Comments to the Author

The revisions have greatly improved the clarity and impact of the paper. The Introduction now does a much better job of covering previous modeling studies and motivating the specific gap this study aims to fill, in the context of existing research. The re-parameterization based on realistic values for SARS-CoV-2 is a big improvement, and these parameters are well explained and justified in the Methods. The Results section is generally very well-written and easy to follow. The narrative explanation of the models described in Figs. 1-2 and the equations has greatly improved the presentation of the Methods. Several more specific questions about the model formulations still remain.

The distinction between latent and infectious individuals is not very clear from page 5. If the two classes are distinguished based on the onset of symptoms, then it makes sense for latent infections to potentially contribute to transmission, as written. But in that case, latent individuals should also test PCR positive (at least sometimes), which contradicts the statement that susceptible and latent individuals would test negative. Distinguishing latency versus infectiousness based on symptom onset is awkward because we know some infectious individuals remain asymptomatic yet infectious throughout their infections (so having an 'infectious' class that these individuals are not a part of would be a misnomer). If the distinction is purely based on symptom presentation, then the classes would be more clearly stated as asymptomatic/symptomatic; otherwise, latent/infectious makes sense if latent individuals are not assumed to contribute to transmission. To me, the cleanest way to distinguish these states is exposed, asymptomatic infectious, and symptomatic infectious (with the potential to skip the latter and go straight to recovered). If the model is intentionally lumping the exposed (non-

infectious) and the a/pre-symptomatic infectious into a single category for simplicity, that is potentially reasonable but needs to be stated more directly. The use of “infectious” only for symptomatic infections seems misleading.

While the text says “traced latently infected individuals are assumed to be sero-negative and therefore receive a negative RNA test at rate kappa...” I do not see the corresponding transition from Lt to Lu at a rate kappa that I would expect. Does this assumption mean that traced latently infected individuals who test negative remain in the traced latent state (i.e., in quarantine if that is in effect) unless they develop symptoms?

In the second model, it appears that the assumption that latent individuals test PCR negative has changed from the first model, because there is no path from Lu to Lc at the rate kappa in Eq. 5f or Fig. 2. If latent individuals test PCR negative, then they should be able to undergo certification just like PCR-negative susceptible individuals (which would be a major potential drawback to certification strategies based on imperfect PCR). If instead we are now assuming that latent individuals would test PCR positive, then why is that assumption different from the first model? I am also very confused as to why the rate of receiving PCR test results (kappa) affects the rate of transition from latent and infectious to recovered classes in the uncertified category. It doesn't make sense that a person's testing status would affect their underlying infection status.

I don't understand the notation used in Eq. 7, because it appears that only latent individuals (Lc and Lu) are contributing to and experiencing new infections, based on the R_XY notation. Is this because individuals never get to the infectious category in this scenario because certification identifies them while still latently infected? I see that this is also the case in Eq. 14 of the appendix, but I don't understand why it arises.

Minor comments:

On page 13, “increases the total number of cases averted approximately tenfold from 250,000 to 1,000,000” is incorrect. (And again on page 19, this claim of a tenfold reduction may now be incorrect with the new parameterization.)

Given how dramatic the result in Figure 8 is (i.e., that suppression can be achieved with generalized interventions and certification testing validity of up to 24 days and waiting time of less than 19 days), I think it is important to restate on page 15 the major assumptions underlying this: that this assumes a very large (78%) reduction in mixing between certified and non-certified people and that, as far as I can tell, certification is only limited by testing turn-around times and not by the proportion of people getting tested. These considerations are in addition to the sharpness of the boundary and the potential sensitivity to other parameters already discussed on page 15.

At the beginning of the Discussion, effective vaccines are now at hand (!!) so this statement could be rephrased to discuss deployment and distribution of effective vaccines. The new variant B.1.1.7 with 50% higher transmissibility could also be discussed in the context of control strategies.

There are several typos in the second sentence of the section “Strategic approaches to suppressing transmission” (page 19).

Decision letter (RSPB-2020-3074.R0)

15-Jan-2021

Dear John,

I am pleased to inform you that your manuscript RSPB-2020-3074 entitled "Five approaches to the suppression of SARS-CoV-2 without intensive social distancing" has been accepted for publication in Proceedings B.

The revised version has been reviewed by the two original referees, one of whom is happy with it as it is, whereas as the second has suggested some minor revisions to your manuscript, in particular requiring clarification on the model formulation. Therefore, I invite you to respond to the referee's comments and revise your manuscript. Because the schedule for publication is very tight, it is a condition of publication that you submit the revised version of your manuscript within 7 days. If you do not think you will be able to meet this date please let us know.

Sincerely,
Professor Loeske Kruuk
mailto:proceedingsb@royalsociety.org

Referee: 1
Comments to the Author(s).
None - Accept As Is.

Referee: 2
Comments to the Author(s).

The revisions have greatly improved the clarity and impact of the paper. The Introduction now does a much better job of covering previous modeling studies and motivating the specific gap this study aims to fill, in the context of existing research. The re-parameterization based on realistic values for SARS-CoV-2 is a big improvement, and these parameters are well explained and justified in the Methods. The Results section is generally very well-written and easy to follow. The narrative explanation of the models described in Figs. 1-2 and the equations has greatly improved the presentation of the Methods. Several more specific questions about the model formulations still remain.

The distinction between latent and infectious individuals is not very clear from page 5. If the two classes are distinguished based on the onset of symptoms, then it makes sense for latent infections to potentially contribute to transmission, as written. But in that case, latent individuals should also test PCR positive (at least sometimes), which contradicts the statement that susceptible and latent individuals would test negative. Distinguishing latency versus infectiousness based on symptom onset is awkward because we know some infectious individuals remain asymptomatic yet infectious throughout their infections (so having an 'infectious' class that these individuals are not a part of would be a misnomer). If the distinction

is purely based on symptom presentation, then the classes would be more clearly stated as asymptomatic/symptomatic; otherwise, latent/infectious makes sense if latent individuals are not assumed to contribute to transmission. To me, the cleanest way to distinguish these states is exposed, asymptomatic infectious, and symptomatic infectious (with the potential to skip the latter and go straight to recovered). If the model is intentionally lumping the exposed (non-infectious) and the a/pre-symptomatic infectious into a single category for simplicity, that is potentially reasonable but needs to be stated more directly. The use of “infectious” only for symptomatic infections seems misleading.

While the text says “traced latently infected individuals are assumed to be sero-negative and therefore receive a negative RNA test at rate κ ...” I do not see the corresponding transition from L_t to L_u at a rate κ that I would expect. Does this assumption mean that traced latently infected individuals who test negative remain in the traced latent state (i.e., in quarantine if that is in effect) unless they develop symptoms?

In the second model, it appears that the assumption that latent individuals test PCR negative has changed from the first model, because there is no path from L_u to L_c at the rate κ in Eq. 5f or Fig. 2. If latent individuals test PCR negative, then they should be able to undergo certification just like PCR-negative susceptible individuals (which would be a major potential drawback to certification strategies based on imperfect PCR). If instead we are now assuming that latent individuals would test PCR positive, then why is that assumption different from the first model? I am also very confused as to why the rate of receiving PCR test results (κ) affects the rate of transition from latent and infectious to recovered classes in the uncertified category. It doesn't make sense that a person's testing status would affect their underlying infection status.

I don't understand the notation used in Eq. 7, because it appears that only latent individuals (L_c and L_u) are contributing to and experiencing new infections, based on the R_{XY} notation. Is this because individuals never get to the infectious category in this scenario because certification identifies them while still latently infected? I see that this is also the case in Eq. 14 of the appendix, but I don't understand why it arises.

Minor comments:

On page 13, “increases the total number of cases averted approximately tenfold from 250,000 to 1,000,000” is incorrect. (And again on page 19, this claim of a tenfold reduction may now be incorrect with the new parameterization.)

Given how dramatic the result in Figure 8 is (i.e., that suppression can be achieved with generalized interventions and certification testing validity of up to 24 days and waiting time of less than 19 days), I think it is important to restate on page 15 the major assumptions underlying this: that this assumes a very large (78%) reduction in mixing between certified and non-certified people and that, as far as I can tell, certification is only limited by testing turn-around times and not by the proportion of people getting tested. These considerations are in addition to the sharpness of the boundary and the potential sensitivity to other parameters already discussed on page 15.

At the beginning of the Discussion, effective vaccines are now at hand (!) so this statement could be rephrased to discuss deployment and distribution of effective vaccines. The new variant B.1.1.7 with 50% higher transmissibility could also be discussed in the context of control strategies.

There are several typos in the second sentence of the section “Strategic approaches to suppressing transmission” (page 19).

Author's Response to Decision Letter for (RSPB-2020-3074.R0)

See Appendix B.

Decision letter (RSPB-2020-3074.R1)

27-Mar-2021

Dear Dr Drake

I am pleased to inform you that your manuscript entitled "Five approaches to the suppression of SARS-CoV-2 without intensive social distancing" has been accepted for publication in Proceedings B.

Data Accessibility section

Open Access

Paper charges

You are allowed to post any version of your manuscript on a personal website, repository or preprint server. However, the work remains under media embargo and you should not discuss it

with the press until the date of publication. Please visit <https://royalsociety.org/journals/ethics-policies/media-embargo> for more information.

Sincerely,
Editor, Proceedings B
<mailto:proceedingsb@royalsociety.org>

Appendix A

UNIVERSITY OF
GEORGIA
Odum School of Ecology

December 9, 2020

Re: Resubmission of *Five approaches to the suppression of SARS-CoV-2 without intensive social distancing*

Dear Editor,

Please find enclosed with this letter our revised manuscript “Five approaches to the suppression of SARS-CoV-2 without intensive social distancing” which we resubmit for publication as a research article in *Proceedings of the Royal Society B*.

We thank the reviewers for identifying areas for improvement in the original manuscript. The comments have been carefully considered. We have addressed specific comments below (our responses in *italics*):

Referee: 1

Comments to the Author(s):

In my opinion, this study is well conducted and its findings are interesting, since they compare five approaches to suppress the SARS-CoV-2 epidemic without intensive social distancing.

We thank the reviewer for their positive and supportive opinion of our work.

Different models and equations used in this study are well described, however their descriptions are not always easy to follow. I would suggest the Authors to improve the paragraphs related to the description of models. This could be useful, especially for readers without a statistical background.

Both reviewers commented on this. Accordingly, the revised manuscript includes lengthier explanations of the model structure (in the paragraphs around the relevant equations) and uses written expression in words in addition to symbols. We hope that this

will make the model structures more accessible to a wide range of readers and clarify ambiguity about our interpretation of model equations.

Moreover, I would suggest to include appropriate references and descriptions of assumed parameters that regulate each model (e.g. transmission parameters, incubation period etc).

Finally, I would ask if the authors considered the proportion of undocumented cases (please consider the following articles DOI: 10.3390/jcm9051350; DOI: 10.1126/science.abb3221). It is possible that different approach might be differentially affected by undocumented cases?

Both reviewers have requested we parameterize the model more specifically for SARS-CoV-2, which we have done in the revised version. Specifically, we have modified some parameters to match published values, reported in new Table 1. References are provided for all of those parameters. Although this re-parameterization changed some of the numerical details, none of our overall conclusions were affected.

We agree that cases have been widely under-reported and have added one of the two references you provided. Our model is not fit to reported cases and so the concern about undocumented cases primarily affects Strategy 1 containment, where documenting infection is part of the intervention process. In this model, the probability of case detection is given by the parameter q . The way in which case detection affects controllability is shown in Figures 4, 5 and 6 (which has q on the x-axis).

Referee: 2

Comments to the Author(s):

This paper uses models to compare five distinct non-pharmaceutical strategies for controlling the COVID-19 pandemic without extreme social distancing (lockdowns). This is an interesting, important, and timely topic. This paper attempts to fill a somewhat narrow niche by using a general model based on SARS-CoV-2, without any specific parameter information, to inform general control strategies, but presenting results that mainly focus on quantitative (or semi-quantitative “ballpark”) results. The general results (e.g., the first sentence of the Conclusions) are predictable without the model. The general conclusions about the need for high rates of case ascertainment and narrow ranges of testing speed and durability are also self-evident.

While we agree that certain conclusions are obvious, we point to other conclusions which are less obvious. For example, Figures 4, 5, and 6 all illustrate that there is a minimal impact of quarantine on suppressing transmission when active case finding and generalized interventions are implemented. We also find surprising the result that the certification process is capable of substantially suppressing transmission even without generalized interventions, though this would require very rapid testing with a short period of validity (less than a week), as illustrated in Figure 7.

The main compelling results rest on semi-quantitative outcomes—for example, that 95% of cases must be ascertained to suppress the epidemic without generalized interventions, and that test results must come back within 3 days and last less than 2 weeks to be effective. But given that the paper does not discuss parameter estimation or couch the parameter values in published or publicly available sources, we can't put much stock in these more quantitative results (even though it is tempting to do so).

We have adopted the reviewer's suggestion. The revised manuscript includes new parameter values reported in a new, fully references Table 1. The reparameterization has, of course, changed the numerical details of output and figures, but the study conclusions remain the same.

As written the Introduction, Discussion, and Conclusions provide insufficient background and context to really inspire broad interest.

The Introduction now makes reference to several other modeling studies and more clearly defines the space in the literature which our study aims to fill. In light of the great volume of recent COVID-19 modeling literature (as noted in the next comment), instead of

providing a lengthy overview of other studies, we subdivided the Introduction into subsections. We believe this helps to more clearly focus and place our study in the broader context of COVID-19 research. The Discussion section has been expanded to provide background and context on non-pharmaceutical approaches to minimizing the impact of COVID-19. We have also made reference to on-the-ground studies of the effectiveness of testing, contact tracing, and household quarantine. The Conclusions section has been expanded to express the significance of our findings and to set out recommendations for COVID-19 management policies in a practical setting.

The Introduction is very brief, and does not include an overview of previous modeling efforts that have addressed similar questions (of which there are many). Though it is difficult for anyone to stay on top of the COVID-19 modeling literature (especially as so much of it currently exists in preprints), it is important for this paper to identify a clear gap that it seeks to fill that other papers have not already considered, and to discuss the rationale for using a generic model not parameterized for any specific location to make inferences about the efficacy of different control strategies. For example, have any previous models evaluated some but not all of these strategies, or considered them individually but not together?

We have included reference to other modeling studies of COVID-19 in the Discussion section. To our knowledge, no other modeling studies have considered the effectiveness of temporary certification for suppressing transmission of SARS-CoV-2. However, we have made reference to studies of durable certification or “shield immunity”/“cocooning” which have been proposed and considered by other researchers. We have expanded the Introduction to reference other modeling studies and have added text to clearly identify the gap in the literature which our work aims to fill.

The equations presented in (1) are hard to follow. Why does the RNA testing rate affect only the transition from S_t to S_u and not from L_t to I_t ? Why do L_t individuals transition into L_u ? More build-up of the model approach would be useful here, going step by step, numbering and referring to the equations individually, and defining all parameters in the text and not just in Fig. 1.

We have added further explanation of the process represented by the equations in (1), explaining the approach step-by-step with reference to individual equations, as recommended. RNA tests are the mechanism by which uninfected (susceptible) traced individuals may move to untraced compartments. Therefore if a latently-infected or infectious traced individual were to receive a (positive) RNA test, they would not transition and remain in their respective traced compartment.

Before diving into the analysis strategy (calculating R_0), the Methods should give an overview of the approach and goals of the models and analysis. Why is R_0 calculated here? Is that the metric of control that the analyses will use? This does not seem well aligned with the goal of studying the dynamics of the epidemic once it has emerged (especially for interventions that can affect the recovered class, such as certification). Or will simulations of dynamics also be used?

We believe that the Overview section preceding Methods provides a sufficient explanation of the goals of the study. However, we agree that the relationships of these goals to the approaches used are not clearly explained. To that end, we have added a paragraph in the Methods section, preceding the descriptions of the strategies, outlining the approaches used for analyzing the models. We describe how both simulations and critical criteria (R_0) are used to explore the effectiveness of each strategy.

In the paragraph describing the interpretation of force of infection functions (page 5), referring to parameters by their biological interpretations and not just by symbols would make this section much easier to interpret. Giving the intuition for why the functions are interpreted as they are is necessary to allow a reader to follow the rationale.

Sentences interpreting the force of infection functions have been added to the relevant sections for both strategies.

In Model 2, why are certified individuals only assumed to enter the S class and not the R class, even though the text discusses “durable” certification based on antibody testing? It doesn’t seem to make sense to consider the effects of certification on R_0 , since a primary goal of (durable) certification is to identify individuals who are immune and can safely interact with others without risk of transmission.

We have added text further describing the two routes of certification and connected them to the parameters and transitions of the model. Notably, durable certification (through the parameter δ) does not affect R_0 as is discussed in Appendix 1. On the other hand, as expressed in the equation following (7), the rate of temporary certification testing (κ) and the period of certification validity ($1/x_i$) can have an impact on R_0 . However, this relationship is highly non-linear.

This focus on R_0 (page 7) leads to the somewhat misleading statement “if certification remains valid indefinitely ... then the risk of an outbreak is the same as if there were no certification process”, even though the focus of this paper is not on the probability of an outbreak but on the dynamics of that outbreak once it occurs. The statement “if certification has no effect ... then there is significantly lower risk of an outbreak” also does not really make sense; it’s not clear

what ‘lower’ is relative to, or how the rate of RNA testing is assumed to affect transmission (via isolation of infected people, presumably, but this isn’t spelled out in the description of the model).

We have modified these statements to no longer use probabilistic language. The relationship between RNA testing and transmission has been further spelled out, as described in our previous comment.

The “Methods: Generalized interventions” sub-section needs more information. Which intervention levels are considered, and why? How does this compare to estimated reductions in R_{eff} that have occurred since the start of the pandemic? Determining reasonable levels of these interventions is the key element to determining the relative effectiveness of these different intervention strategies.

We agree that the effect of these interventions is important to determining the impact of the strategies that we consider. To that end, we have added references to studies which have attempted to measure the effect of generalized interventions on transmission in “Methods: Generalized interventions”. While these measurements vary, our assumed value falls within the reasonable range. We also added a few sentences highlighting other modeling studies which have considered this level of effect (50%).

Under the Implementation section, why was pomp used? Is aspect of the model stochastic? Were there any parameters estimated, from any data? Again, we need better context about what purpose the analyses served and what their main approaches were, before diving into the details.

The package ‘pomp’ provides functions for both estimation and solution. We have used it here for deterministic solutions with the intention of extending these models to stochastic solution and estimation problems in the future.

How were the parameter values chosen? There is plenty of parameter information publicly available (e.g., through the MIDAS network COVID-19 modeling portal), so further justification of parameter choices is needed. Without knowing how realistic the parameter values are, it is unclear how this model can inform the COVID-19 response.

As described above in our response to Reviewer 1, we have modified some parameters to match published values, reported in new Table 1. References are provided for all of those parameters. Although this re-parameterization changed some of the numerical details, none of our overall conclusions were affected.

In the second paragraph of the Results section, what are the “generalized interventions” used here? What does this mean in terms of parameter values and in terms of interpretation (e.g., mask-wearing, physical distancing, etc.)? Since generalized interventions are important throughout most of the results, it is important to revisit what is assumed here.

We have expanded the “Methods: Generalized interventions” subsection prior to this section to clarify how generalized interventions are interpreted in terms of changes in parameter values. This sentence in the second paragraph of Results has been changed to remind readers what is meant by generalized interventions.

In the Results, the choice of parameter values becomes important in making statements like the last sentence on page 8 (95% of cases have to be identified to achieve suppression). This number is likely to be sensitive to the choice of transmission rates, recovery rates, etc. It would also be helpful (in addition to Fig. 4) to see simulated dynamics for these intervention scenarios (as incident cases, rather than cumulative cases), so we can see how interventions affect the epidemic peak and how long they need to be maintained for the epidemic to burn out completely (if ever).

The number referenced in this comment, the critical case finding rate (q^), is described in detail in Methods: Strategy 1. The formula for how it is computed as a function of the other transmission parameters is also included. In Appendix 2, we provide context for the sensitivity of outputs to transmission parameters. We have also added plots of incident cases over time for each intervention scenario to Appendix 2.*

On page 10, what is the distinction between contact tracing and quarantine? This is worth restating here.

We added two sentences to remind readers and to clarify the distinction between these. Specifically,

“When untraced individuals are identified through active case finding or contact tracing, they are then traced and isolated, reducing their ability to transmit. On the other hand, traced infectious individuals in quarantine do not contribute at all to transmission.”

For the “When can certification be effective?” results, how dependent is this on the proportion of people who can receive testing (undergo possible certification)? Is there a tradeoff between proportional testing coverage and testing validity and wait times?

While the overall population is subdivided into certified and uncertified subpopulations, those in the uncertified susceptible subpopulation are considered to have access to testing at the same rate. We have added further clarification of this to the description of Strategy 2 in Methods. We do not assume a tradeoff between testing validity period and test waiting times.

The Discussion section is insufficient. It needs a general overview of the key results and their implications for COVID-19 control.

We have expanded the Discussion section “Strategic approaches to suppressing transmission” to include an overview of key results. The Conclusions have been expanded to frame these results and their implications in terms of COVID-19 control.

It needs to place the results in context with other COVID-19 literature.

We have added text relating our results to previous literature on SARS-CoV-2 transmission to the beginning of the Discussion section, preceding “Are our parameters realistic for COVID-19?”.

It needs to discuss possible model sensitivity to parameters and other assumptions. For example, the section “Are our parameters realistic for COVID-19?” only discusses the asymptomatic fraction, and does not discuss any other important parameters that are likely to affect the results, such as transmission rates, recovery rates, mixing rates, and relative infectiousness rates.

It needs to discuss how different intervention types have been applied already, and how successful they have been.

We added text to the discussion to address this. While there seem to be many modeling studies, and estimates of the impact of population-wide general NPI, analyses that quantify the impact of specific interventions (e.g. a screening/testing campaign) seem to not currently exist.

Minor comments:

Introduction and Discussion: “represented by incommensurable flow diagrams” is an opaque way of phrasing this.

We have rephrased this to be clearer to the reader, specifically we now state that they are “structurally different”. We believe this sufficiently elucidates our meaning for the general reader. For the mathematically inclined reader, a comparison of the subsequent equations illuminates the sense in which they are structurally different (i.e. involving both different flows and different state variable definitions).

In the Model 2 methods on page 5, the first sentence following equations (4) is unclear (both certified and uncertified persons who are S, L, and I, but two pools of R).

This sentence has been modified for clarity. (Page 7, first sentence following equations)

The distinction between Figs. 4 and 5 should be made clearer in the caption.

Figure captions have been expanded throughout the manuscript to clarify the differences among related graphs. The caption of Figure 5 now explains that perfect isolation means no transmission from traced latent and infectious individuals.

Page 11 “by comparing the curves in Figures 4 and 5” is not a very effective way to present results, given that the two figures look almost identical but with different y-axis scaling. Isn't the difference between them what's plotted in Fig. 6, a much better visualization of this result?

We agree and have pointed readers to Fig. 6 which better illustrates the point made.

The Appendices need to be written out with equation numbers and figure numbers, as in the main text and these figures need to be referred to in the main text. There is not enough context in Appendix 2 to interpret these figures.

This edit has been made. Captions for figures in Appendix 2 have been expanded to provide context for their interpretations.

We also made these minor changes to the text to improve clarity and context

To improve clarity, the revised manuscript makes a distinction between the basic and control reproduction numbers

We have also added text to Methods: Strategy 1 clarifying the sensitivity of the critical case-finding proportion (q^) to the transmission parameters for the traced infectious (bIt) and untraced latently-infected (bLu) compartments.*

These results are original and not under consideration for publication elsewhere. All authors have approved the attached manuscript for submission.

I can be reached by email (jdrake@uga.edu) or phone (+1 706-818-4452). Thank you for your consideration.

Sincerely,

John M. Drake
University of Georgia

Appendix B

UNIVERSITY OF
GEORGIA
Odum School of Ecology

January 20, 2021

Re: Resubmission of *Five approaches to the suppression of SARS-CoV-2 without intensive social distancing*

Dear Editor,

Please find enclosed with this letter our revised manuscript “Five approaches to the suppression of SARS-CoV-2 without intensive social distancing” which we resubmit for publication as a research article in *Proceedings of the Royal Society B*.

We thank the reviewers for identifying areas for improvement in the original manuscript. The comments have been carefully considered. We have addressed specific comments below (our responses in *italics*):

Referee: 1

Comments to the Author(s).

None - Accept As Is.

Referee: 2

Comments to the Author(s).

The revisions have greatly improved the clarity and impact of the paper. The Introduction now does a much better job of covering previous modeling studies and motivating the specific gap this study aims to fill, in the context of existing research. The re-parameterization based on realistic values for SARS-CoV-2 is a big improvement, and these parameters are well explained and justified in the Methods. The Results section is generally very well-written and easy to follow. The narrative explanation of the models described in Figs. 1-2 and the equations has greatly improved the presentation of the Methods. Several more specific questions about the model formulations still remain.

We thank the reviewer for their thoughtful and supportive remarks. Responses to those specific questions follow below the reviewer's comments.

The distinction between latent and infectious individuals is not very clear from page 5. If the two classes are distinguished based on the onset of symptoms, then it makes sense for latent infections to potentially contribute to transmission, as written. But in that case, latent individuals should also test PCR positive (at least sometimes), which contradicts the statement that susceptible and latent individuals would test negative. Distinguishing latency versus infectiousness based on symptom onset is awkward because we know some infectious individuals remain asymptomatic yet infectious throughout their infections (so having an ‘infectious’ class that these individuals are not a part of would be a misnomer). If the distinction is purely based on symptom presentation, then the classes would be more clearly stated as asymptomatic/symptomatic; otherwise, latent/infectious makes sense if latent individuals are not assumed to contribute to transmission. To me, the cleanest way to distinguish these states is exposed, asymptomatic infectious, and symptomatic infectious (with the potential to skip the latter and go straight to recovered). If the model is intentionally lumping the exposed (non-infectious) and the a/pre-symptomatic infectious into a single category for simplicity, that is potentially reasonable but needs to be stated more directly. The use of “infectious” only for symptomatic infections seems misleading.

We agree that this distinction can be made more clear in the text. We assume that individuals with a latent or incubating infection are infectious. Asymptomatic and symptomatic infectious individuals are considered to be in the “infectious” compartments of the two models. We make the assumption that asymptomatic individuals are capable of transmitting the virus onwards, reduced by a factor of b_L . However, asymptomatic individuals are less likely to seek out testing and their lower viremia may not be detected by a test (though test sensitivity has certainly increased over time). Thus, as a conservative estimate, we assume the case detection rate (q) to be 50%, corresponding to the CDC’s estimate of the upper bound on the percentage of cases which are asymptomatic.

We have adjusted the descriptions of these epidemiological compartments to align with these suggestions. We no longer apply the term “pre-symptomatic” to latent infections as this can imply that all incubating infections eventually become symptomatic, which is not an assumption of our model. Please see page 5, paragraph 1 and Table 1.

While the text says “traced latently infected individuals are assumed to be sero-negative and therefore receive a negative RNA test at rate κ ...” I do not see the corresponding transition from L_t to L_u at a rate κ that I would expect. Does this assumption mean that traced latently infected individuals who test negative remain in the traced latent state (i.e., in quarantine if that is in effect) unless they develop symptoms?

That transition was formerly included in the model but then removed because we shared the concerns brought up here. We intended to remove that sentence in the previous draft and have removed it in this current revision.

In the second model, it appears that the assumption that latent individuals test PCR negative has changed from the first model, because there is no path from Lu to Lc at the rate kappa in Eq. 5f or Fig. 2. If latent individuals test PCR negative, then they should be able to undergo certification just like PCR-negative susceptible individuals (which would be a major potential drawback to certification strategies based on imperfect PCR). If instead we are now assuming that latent individuals would test PCR positive, then why is that assumption different from the first model? I am also very confused as to why the rate of receiving PCR test results (kappa) affects the rate of transition from latent and infectious to recovered classes in the uncertified category. It doesn't make sense that a person's testing status would affect their underlying infection status.

We have removed the erroneous sentence in the description of the first model noted above. There is no path from Lu to Lc in Strategy 2 (or Lt to Lu for Strategy 1).

In Strategy 1, RNA testing is not used to implement isolation of cases. Tracing is only implemented if individuals are identified through contact tracing (alpha) or passive and active case identification (q). However, in Strategy 2 RNA testing is used to identify infectious cases in order to enforce isolation protocols to ensure these individuals no longer contribute new cases. We have further clarified the differences between the strategies in the description of Strategy 2 (see page 6, last paragraph).

The class "R" represents both recovered and removed individuals. We have adjusted the model descriptions to reinforce that individuals in "R" may be either recovered (no longer infected) or removed (incapable of transmission due to intervention).

I don't understand the notation used in Eq. 7, because it appears that only latent individuals (Lc and Lu) are contributing to and experiencing new infections, based on the R_XY notation. Is this because individuals never get to the infectious category in this scenario because certification identifies them while still latently infected? I see that this is also the case in Eq. 14 of the appendix, but I don't understand why it arises.

Based on the form of our model, new infections only occur in the "latent" compartments. A latent infectious individual in a completely susceptible population contributes to new infections in other compartments as they progress through the flow diagram. They may proceed directly to "I" or move to certified/uncertified or traced/untraced, and so on all the while contributing to new latent cases (certified or uncertified). For simplicity of

notation, we opted to define the type reproduction numbers in terms of the compartment of the initial infection and the compartment where new infections are induced. To avoid confusion, we have added a line to clarify why these quantities reference only the latent compartments after Equation 7.

Typographical errors in the expressions for the type reproduction numbers in Equation 14, Appendix 1 have also been fixed.

Minor comments:

On page 13, “increases the total number of cases averted approximately tenfold from 250,000 to 1,000,000” is incorrect. (And again on page 19, this claim of a tenfold reduction may now be incorrect with the new parameterization.)

We intended this sentence to reference the range of cases averted due to perfect isolation as the fraction of cases identified and isolated (q) varies as seen in Figure 6. We have edited the sentence to provide further context: “...as the fraction of cases identified and isolated (q) varies from 25% to 75%...” in the paragraph following Figure 5. The tenfold reduction referenced on page 19, paragraph 4, last sentence, refers to Figure 8 and we have added a direct reference to the figure there for clarity.

Given how dramatic the result in Figure 8 is (i.e., that suppression can be achieved with generalized interventions and certification testing validity of up to 24 days and waiting time of less than 19 days), I think it is important to restate on page 15 the major assumptions underlying this: that this assumes a very large (78%) reduction in mixing between certified and non-certified people and that, as far as I can tell, certification is only limited by testing turn-around times and not by the proportion of people getting tested. These considerations are in addition to the sharpness of the boundary and the potential sensitivity to other parameters already discussed on page 15.

We agree that this is an important caveat. We have added a sentence clarifying that this result is dependent on a substantial reduction in mixing due to certification (page 15, top of page, last sentence).

At the beginning of the Discussion, effective vaccines are now at hand (!!) so this statement could be rephrased to discuss deployment and distribution of effective vaccines. The new variant B.1.1.7 with 50% higher transmissibility could also be discussed in the context of control strategies.

We have rephrased the first sentence of the discussion to refer to vaccines currently in distribution and the new, more transmissible variant of SARS-CoV-2. We appreciate this comment which helped the authors to better connect our work to these new developments.

There are several typos in the second sentence of the section “Strategic approaches to suppressing transmission” (page 19).

We have corrected the typos and grammar in this sentence. The authors would like to reiterate their gratitude towards the reviewer for their close reading of our work.

These results are original and not under consideration for publication elsewhere. All authors have approved the attached manuscript for submission.

I can be reached by email (jdrake@uga.edu) or phone (+1 706-818-4452). Thank you for your consideration.

Sincerely,

John M. Drake
University of Georgia